# Diapycnal dissolved organic matter supply into the upper Peruvian oxycline

Alexandra N. Loginova[1], Sören Thomsen[1,2], Marcus Dengler[1], Jan Lüdke[1], Anja Engel[1*]

[1]GEOMAR Helmholtz-Centre for Ocean Research Kiel, Düsternbrooker Weg 20, Kiel, 24105, Germany

[2]LOCEAN-IPSL, IRD/CNRS/Sorbonnes Universites (UPMC)/MNHN, Paris, UMR 7159, France

*Correspondence to*: Anja Engel (aengel@geomar.de)

**Abstract.** The Eastern Tropical South Pacific (ETSP) hosts the Peruvian upwelling system, which represents one of the most productive areas in the world ocean. High primary production followed by rapid heterotrophic utilization of organic matter supports the formation of one of the most intense oxygen minimum zones (OMZ) in the world ocean, where dissolved

oxygen ($O_2$) concentrations reach well below 1 µmol kg$^{-1}$. The high productivity leads to an accumulation of dissolved organic matter (DOM) in the surface layers that may serve as a substrate for heterotrophic respiration. However, the importance of DOM utilization for $O_2$ respiration in the Peruvian upwelling system in general and for shaping the upper oxycline in particular remains unclear so far. This study reports the first estimates of diapycnal fluxes and supply of $O_2$, dissolved organic carbon (DOC), dissolved organic nitrogen, dissolved hydrolysable amino acids (DHAA) and dissolved

combined carbohydrates (DCCHO) for the ETSP off Peru. Diapycnal flux and supply estimates were obtained by combining measured vertical diffusivities and solute concentration gradients. They were analysed together with the molecular composition of DCCHO and DHAA to infer the transport of labile DOM into the upper OMZ and the potential role of DOM utilization for the attenuation of the diapycnal $O_2$ flux that ventilates the OMZ. The observed diapycnal $O_2$ flux (50 mmol $O_2$ m$^{-2}$ day$^{-1}$ at max) was limited to the upper 80 m of the water column, the $O_2$ supply of ~1 µmol kg$^{-1}$ day$^{-1}$, was comparable to

previously published $O_2$ consumption rates for the North and South Pacific OMZs. The diapycnal DOM flux (31mmol C m$^{-2}$ day$^{-1}$ at max) was limited to ~30 m water depth, suggesting that the labile DOM is extensively consumed within the upper part of the shallow oxycline off Peru. The analyses of DCCHO and DHAA composition support this finding, suggesting that DOM undergoes comprehensive remineralization already within the upper part of the oxycline, as the DOM within the core of the OMZ was found to be largely altered. Estimated by a simple equation for carbon combustion, aerobic respiration of

DCCHO and DHAA, supplied by diapycnal mixing (0.46 µmol kg$^{-1}$ day$^{-1}$ at max), could account for up to 38% of the diapycnal $O_2$ supply in the upper oxycline, which suggests that DOM utilization plays a significant role for shaping of the upper oxycline in the ETSP.

# 1 Introduction

Dissolved oxygen ($O_2$) plays a key role for biological production and cycling of elements in marine ecosystems as well as for the spatial distribution of marine organisms (Ekau et al., 2010, Gilly et al., 2013). The majority of catabolic processes in organisms are conducted by oxidation with $O_2$ (e.g. Bender and Heggie 1984). The eastern tropical South Pacific (ETSP) embodies one of the largest oxygen minimum zones (OMZ) in the world ocean (Karstensen et al., 2008; Paulmier and Ruiz-Pino, 2009). The core of the Peruvian OMZ is considered to be fully anoxic (e.g. Ulloa et al. 2012), as $O_2$ concentrations below the detection limit (DL) of ~0.01 µmol $kg^{-1}$ were observed between 20 and 400 m depth by high precision STOX sensor measurements (Revsbech et al., 2009; Kalvelage et al., 2013; Thomsen et al., 2016a). Those low $O_2$ concentrations are due to a sluggish ventilation by ocean currents, carrying low-$O_2$ waters to the ETSP and microbial respiration attributed to utilization of organic matter (OM) originating from the upper water column (e.g. Czeschel et al., 2011; Brandt et al., 2015; Kalvelage et al., 2015).

Elevated primary production in the Peruvian upwelling region above the OMZ (Pennington et al., 2006) leads to an accumulation of both particulate (POM) (Franz et al., 2012a) and dissolved (DOM) organic matter (Romankevich and Ljutsarev 1990; Franz et al., 2012a; Letscher et al., 2013; Loginova et al., 2016) in the euphotic zone at the continental margin. POM was recognized to be an important source of carbon (C) for microbial OM mineralization (e.g. Dale et al., 2015), utilization of $O_2$ (Kalvelage et al.,2015), and anaerobic processes, such as nitrogen (N) loss via denitrification (Kalvelage et al., 2013; Chang et al., 2014), in the area. However, the cycling of DOM in the Peruvian upwelling system has been little studied.

DOM, which originates in the euphotic zone, as a result of extracellular release by phytoplankton, cell lysis, particle degradation and sloppy zooplankton feeding (Benner, 2002), is commonly enriched in labile and semi-labile DOM. Those are mainly composed of carbohydrates (CHO) and amino acids (AA) (e.g. Ogawa and Tanoue, 2003). CHO and AA are preferentially utilized during microbial decomposition of OM, as they serve as energy sources and "building blocks" for microbes to respire and grow (Skoog and Benner, 1997; Lee et al., 2000; Amon et al., 2001). Thus, the rapid microbial decomposition of labile organic matter in the euphotic zone is commonly followed by slower decomposition of less bioavailable semi-labile DOM and very slow decomposition of extensively reworked refractory DOM deeper the water column (e.g. Hansell, 2013). Therefore, the composition of DOM reflects its diagenetic history, and the contribution of CHO and AA to DOM, may be used as a measure of DOM bioavailability (Davis et al, 2007; 2009; Kaiser and Benner, 2009).

Microbial decomposition of organic matter has previously been suggested to be limited under anoxia (Harvey et al., 1995; Nguyen and Harvey, 1997). Following this suggestion, one may assume that, if labile DOM is supported to the OMZ, it would not be reworked as rapidly as in oxygenated waters. Recent studies in the upwelling area and the corresponding OMZ off Chile found, however, that even under anoxia the ability of microbes to decompose labile DOM (Leucine-incorporation rate) did not differ from the oxygenated waters (Sempéré et al., 2008, Pantoja et al., 2009). These studies suggest that slower remineralization of DOM in OMZ might rather be caused by lack of bioavailable organic matter supply into the OMZ than by low-$O_2$ conditions. Herewith, measured concentrations of bioavailable components of DOM over the water column in the ETSP are yet controversary. For instance, Pantoja et al. (2009) reported relatively high concentrations of free and combined

AA in the OMZ off Chile. Sempere et al. (2008) reported low concentrations of neutral CHO in the corresponding upwelling area, compared to the open Pacific Ocean.

Contrary to POM, DOM does not obtain its own buoyancy and its transport is exclusively due to advective and diffusive physical transport processes (e.g. Löscher et al. 2016). In upwelling regimes, turbulent mixing processes are often enhanced at the continental margin resulting in high diapycnal fluxes of various solutes (e.g. Schafstall et al., 2010; Kock et al., 2012; Brandt et al., 2015; Steinfeldt et al., 2015). On the other hand, the downward fluxes of DOM, or other solutes, may be

reduced or even predominated by upwelling fluxes due to Ekman divergence in the coastal upwelling region (e.g. Steinfeldt et al., 2015). Machadevan (2014) suggested that transport of OM (via eddy fluxes) into the OMZ should be accompanied by $O_2$ in amount that is sufficient for full remineralization of the subducted OM. Therefore, this physical transport of OM and $O_2$ should stimulate heterotrophic aerobic respiration in the OMZ, which was suggested to be the main pathway of OM remineralization in the upper OMZs by Kalvelage et al. (2015). However, so far, no direct $O_2$ and DOM supply estimates

exist for the Peruvian OMZ.

Here, we combined physical and biogeochemical observational data that were collected during the R/V METEOR "M93" (M93) research cruise to the ETSP off Peru in February-March 2013. Specifically, we directly estimated the diapycnal $O_2$ and DOM supply into the upper oxycline off Peru. Additionally, we analyzed diapycnal fluxes and the composition of dissolved combined carbohydrates (DCCHO) and dissolved hydrolysable amino acids (DHAA) to learn whether DOM and

its labile and semi-labile constituents may be supplied to the upper OMZ and the potential contribution of DOM based respiration to $O_2$ flux attenuation.

## 2 Methods

### 2.1 Study area

The observational data were acquired during the research cruise "M93" which took place from 7th of February to 9th of

March 2013 between 12°S and 14°S and 76°W and 79°W off Peru (Fig. 1). During the measurement program, the study area was affected by moderate southeasterly winds (1-9 m/s) (Thomsen et al., 2016a). The water column was highly stratified during the cruise (Fig. 2a,b). High concentrations of inorganic nutrients (~30 µmol L$^{-1}$ ($NO_3^-$), ~3 µmol L$^{-1}$ ($PO_4^{3-}$)) just below the surface (Thomsen et al., 2016a) collocated with highest chlorophyll $a$ (chl $a$) concentrations near the surface (5-80 m depth; Fig. 2c) (Loginova et al., 2016). The oxycline was located at upper 5-80 m depth, where oxygen concentrations

dropped from >200 µmol kg$^{-1}$ to <1µmol kg$^{-1}$ (Fig. 2d) (Thomsen et al., 2016a). In summary, our observations were carried out during a period which corresponds to typical summer conditions off Peru.

## 2.2 Discrete water sampling and analyses

Seawater was sampled with a rosette (GO; General Oceanics, USA) equipped with a conductivity, temperature and depth profiler (CTD; Sea-Bird (SBE) 9-plus, Sea-Bird Electronics Inc., USA), an $O_2$ optode (SBE43, Sea-Bird Electronics Inc., USA), a WETStar chl *a* fluorometer (WET Labs, USA) and 24 x 10 L Niskin bottles. Additional water samples were taken

with a PUMP-CTD-System (an integrated measurement device, which was developed in collaboration between Leibniz Institute for Baltic Research (IOW) and the Max Planck Institute for Marine Microbiology (MPI) Bremen: PUMP-CTD; Strady et al., 2008). In general, samples were collected at 3 to 8 sampling depths from 2 to 70 m at the onshore stations (~10km offshore) and from 2 to 200 m at stations offshore (~100 km offshore). DOC/DON analyses were performed for 49 CTD stations, and for 8 PUMP-CTD stations. DHAA and DCCHO analyses were performed only for samples from the GO

rosette. CTD, $O_2$ and chl *a* recordings were taken at 172 profiles (Fig. 1a).

The CTD was calibrated with discrete seawater samples measured with a Guildline Autosal 8 model 8400B salinometer. The $O_2$ optode was calibrated by Winkler titration above the oxycline (Winkler, 1888; Hansen, 1999). The STOX sensor measurements, which revealed $O_2$ concentrations of 0.01-0.05 µmol kg$^{-1}$ within the OMZ (Revsbech et al., 2009; Thomsen et al., 2016a), were used for $O_2$ optode calibration at low $O_2$ levels. The salinity and $O_2$ measurements had precision of 0.002 g

kg$^{-1}$ and ~ 1 µmol kg$^{-1}$, respectively. More details on the salinity and $O_2$ calibrations can be found in Thomsen et al. (2016a). Apparent oxygen utilization (AOU) was then calculated as a difference of measured $O_2$ concentrations and its equilibrium saturation using Gibbs-Sea Water Oceanographic Toolbox (McDougall and Barker, 2011) for MatLab (MathWorks, USA) for analyses of potential relationship between DOM reworking and the utilization of $O_2$.

The original fluorometer calibration provided by the sensor manufacturer (WET Labs, USA) was used throughout the cruise

resulting in chl *a* concentrations in µg L$^{-1}$. More detail on the recalibration of the chl *a* fluorimeter one can find in Loginova et al. (2016).

Net primary production (NPP) was estimated for study area off Peru (12°S-14°S and 76°W-79°W) and the corresponding time period (February 2013) after the model of Behrenfeld and Falkowski (1997a) with Ocean Productivity toolbox (Oregon State University).

DOC/DON duplicate samples (20 mL) were collected into combusted glass ampoules (8 h, 450° C) after filtration with combusted GF/F filters (5 h, 450°C). Samples were acidified (80mL of 85% $H_3PO_4$), sealed with flame and stored at 4°C in the dark until analysis. DOC samples were analysed by the high-temperature catalytic oxidation method (TOC -VCSH, Shimadzu) modified from Sugimura and Suzuki (1988). The detection limit (DL) was 1 µmol L$^{-1}$. Total dissolved nitrogen (TDN) was determined simultaneously to DOC with DL of 2 µmol L$^{-1}$ using the TNM-1 detector of a Shimadzu analyser

[Dickson et al., 2007]. DON concentrations were calculated by subtracting inorganic nitrogen concentrations from concentrations of TDN. The description of the instrument calibration and measurements may be found in Loginova et al. (2015).

Duplicate samples (~16ml) for DCCHO were collected into combusted (8hrs, 450°C) 25ml-glass vials after passing through 0.45 µm syringe filters (GHP membrane, Acrodisk, Pall Corporation) and immediately frozen at -20°C until analyses. Analyses were conducted by high performance anion exchange chromatography (HPAEC) coupled with pulsed amperometric detection following Engel and Händel (2011). Prior to analyses samples were thawed at room temperature and

desalinated by membrane dialysis (1 kDa MWCO, Spectra Por, 5 h at 1°C). Desalinated duplicate subsamples (2 mL) were hydrolyzed using 1.6mL of 1M HCl (for each) for 20 h at 100°C. The hydrolyzed samples were neutralized through acid evaporation under $N_2$ atmosphere and an addition of miliQ water (20mL). DCCHO monomers were determined from 17.5 mL subsamples on a Dionex ICS 3000 system. More detailed method and calibration descriptions are given in Engel and Händel (2011). The method precision was 2% with a DL ~10 nmol $L^{-1}$. During our study, three classes of polysaccharides

were measured. Those were neutral sugars (fucose (Fuc), rhamnose (Rha), arabinose (Ara), galactose (Gal), glucose (Glc), mannose (Man) and xylose (Xyl)), amino sugars (glucosamine (GlcN) and galactosamine (GalN)), and acidic sugars including gluconic acid (GluA)) and the uronic acids galacturonic acid (GalUA) and glucuronic acid (GlcUA). Man and Xyl were quantified as a mixture due to co-elution, and, therefore, reported together (ManXyl). Concentrations of DCCHO after hydrolysis are given as monomer equivalents.

Duplicate samples (~3ml) for DHAA were filtered with 0.45 µm syringe filters (GHP membrane, Acrodisk, Pall Corporation) and stored frozen (-20°C) in combusted (8hrs, 450°C) 4ml-glass vials until analyses. Samples were thawed and hydrolyzed with 6 N HCl at 100°C for 20 h prior to analysis. DHAA were determined by HPLC after ortho-phthaldialdehyde derivatization (Lindroth and Mopper, 1979; Dittmar et al., 2009) with DL of 2 nmol $L^{-1}$ and precision of <5%. The following amino acids were analyzed during the study: α-amino acids: aspartic acid (Asp), glutamic acid (Glu), serine (Ser), arginine

(Arg), glycine (Gly), threonine (Thr), alanine (Ala), tyrosine (Tyr), valine (Val), phenylalanine (Phe), isoleucine (Ileu), leucine (Leu) and γ-amino acid: γ-aminobutyric acid (GABA). The amino acids asparagine and glutamine likely contributed to the measured Asp and Glu concentrations, respectively, due to deamination during hydrolysis. Alpha aminobutyric acid was used as an internal standard to account for losses during handling. Concentrations of DHAA after hydrolysis are given as monomer equivalents. More in-detail description of the method may be found in (Engel and Galgani, 2016).

**2.3 Diapycnal flux calculations**

To estimate the diapycnal fluxes of various solutes, CTD sensor ($O_2$) and bottle data (DOC, DON, DCCHO and DHAA) were combined with near-simultaneous measurements of turbulence in the water column. The turbulence measurements were performed with a microstructure profiling system (MSS) from the rear of the vessel. The loosely-tethered profiler (MSS90-D, S/N 32, Sea & Sun Technology) was optimized to sink at a rate of 0.55 m $s^{-1}$ and was equipped with three shear sensors

and a fast-response temperature recorder, as well as an acceleration sensor, two tilt sensors and CTD, sampling with lower response time.  At each CTD station, 3-6 microstructure profiles were collected. Standard processing procedures were used to determine the rate of kinetic energy dissipation of turbulence in the water column ($\varepsilon$, $m^2s^{-3}$), as given in Schafstall et al. (2010).

Diapycnal diffusivities ($K_\rho$, $m^2 s^{-1}$) were determined at 14 m depth intervals, following Osborn (1980):

$$K_\rho = \Gamma \frac{\varepsilon}{N^2}, \tag{1}$$

where $N$ is stratification (in $s^{-1}$) and $\Gamma$ is the mixing efficiency, for a which value of 0.2 was used. The diapycnal diffusivity of the solutes ($O_2$, DOC, DON, DCCHO, and DHAA) - $K_S$ – was assumed to be equivalent to the diapycnal diffusivity of the mass $K_\rho$ (e.g. Schafstall et al., 2010; Fischer et al., 2013).

5 The diapycnal fluxes (mmol $m^{-2}$ $day^{-1}$) of the different solutes listed above were estimated using Eq. 2, implicitly assuming equivalency of vertical and diapycnal diffusivities ($K_s \approx K_\rho$).

$$\Phi_S = -K_\rho \nabla C_S, \tag{2}$$

where $\nabla C_S$ is the vertical gradient of the molar concentration of the solutes (mmol $m^{-4}$).

The mean diapycnal supply ($-\overline{\nabla \Phi_s}$, $\mu$mol $kg^{-1}$ $day^{-1}$) of a solute was determined at 28 m depth intervals as an attenuation of the diapycnal solute flux profile over depth, according to the Eq. 3:

$$-\overline{\nabla \Phi_s} = -\frac{1}{\rho} \frac{\partial}{\partial z} \overline{\Phi_S}, \tag{3}$$

10 where $\rho$ – is the *in-situ* density of the seawater (kg $m^{-3}$), z - is depth (m) and $\overline{\Phi_S}$ (mmol $m^{-2}$ $day^{-1}$) – is the estimated mean diapycnal flux profile of a solute. The mean diapycnal solute supply was interpreted to balance the amount of a solute that is lost per unit of time over a specific depth interval of the water column due to the microbial utilization of the solute. This interpretation assumes that sources other than turbulent mixing or sinks other than microbial consumption are negligible.

For DCCHO and DHAA the diapycnal flux estimates were based on 14 combined CTD/MSS stations, while for DOC and 15 DON fluxes 22 stations were available (Fig. 1b). The diapycnal $O_2$ flux was determined from 50 combined stations. All combined data sets include stations from the continental slope, as well as stations in deeper waters, where bottom depth was larger than 4000m.

For each combined CTD/MSS station a mean $K_\rho$ was estimated based on a $N^2$ profile (CTD) and mean dissipation profile (turbulence probe) averaged over all MSS profiles conducted at the CTD station. In combination with the vertical solute 20 gradient, a mean flux profile for each station was estimated. Only measurements below the mixed layer, which was defined by a threshold criterion of a 0.2°C temperature decrease below the maximum and a minimum depth of 10 m, were used. Measurements from different sensors and instruments were averaged in temperature space to reduce the impact of internal waves.

The mean diapycnal flux ($\overline{\Phi}_S$) was determined by arithmetically averaging all fluxes from individual stations in 14m depth 25 intervals. The diapycnal solute supply was then determined from the divergence of the mean diapycnal flux ($\overline{\nabla \Phi_S}$).

The 95% confidence interval of the diapycnal flux was calculated following the procedure described by Schafstall et al. (2010). From this error estimate the uncertainty of the supply was derived by error propagation.

A simple equation of carbon combustion:

$$1C + 1O_2 = 1CO_2,$$ (4)

was used for a rough estimation of the percentage of diapycnal $O_2$ supply that may be consumed by heterotrophic communities, if they use all the C, supplied by the diapycnal fluxes of DOC, DCCHO and DHAA.

### 2.4 Statistical analyses of DOM composition

Principal component analysis (PCA) was performed using environmental factors (temperature, salinity and AOU) and relative abundances of $\alpha$-DHAA and neutral DCCHO (mol%) to examine "compositional trends" (i.e. changes in composition in response to an influence of an environmental parameter) in marine DOM in the studied area. The aim of the PCA was also to explore the potential interrelation between low-$O_2$ and DOM composition. For this, temperature, salinity and AOU and relative abundances of labile organic matter from open Atlantic and Pacific Oceans (Kaiser and Benner, 2009) were included in the PCA for the representation of well oxygenated water column. The covariance between principle components and an individual parameter was considered significant when module of the coordinate of the parameter exceeded 0.5 on the "variables factor map". The PCA was performed using "FactorMineR" package (Husson et al., 2010) for "R" (R Core Team, 2013).

### 3 Results

### 3.1 Distribution of $O_2$ and DOM

In this section the horizontal and vertical distribution of $O_2$ and the different DOM components including DOC, DON and their labile and semi-labile constituents, DCCHO and DHAA are described. The vertical gradients of the different solutes are crucial for estimating the associated diapycnal fluxes, as described in section 3.2. Near surface $O_2$ concentrations were observed ranging between 100 µmol kg$^{-1}$ at the coast and 240 µmol kg$^{-1}$ further offshore (Fig. 2d). These values dropped to less than 1 µmol kg$^{-1}$ at <50m depth near the coast (<40 km offshore) and ~80 m depth offshore (>40 km) (Fig. 2d). DOC concentrations ranged from more than 100 µmol L$^{-1}$ near the surface to < 50 µmol L$^{-1}$ below 40 m depth (Fig. 3a). Patches of isolated DOC maxima (up to 120 µmol L$^{-1}$) were measured at a depth range from 20 to 120 m (Fig. 3a). DOC concentrations of >100 µmol L$^{-1}$ had been reported previously for the water column off Peru (Romankevich and Ljutsarev, 1990; Franz et al., 2012a). However, since concentrations >100 µmol L$^{-1}$ were observed only sporadically, we cannot exclude a possible contamination of these samples. The main decrease of DOC occurred between 5 and 30 m. Thus, the main vertical DOC gradient was found at shallow depth, compared to the oxycline. This becomes even more apparent, when comparing the mean vertical profiles of $O_2$ and DOC (Fig. 4a,b).

DON concentrations were also highest (~7-8 µmol L$^{-1}$) near the surface (Fig. 3b) and varied from below detection to 4-5 µmol L$^{-1}$ at greater depth. The main decrease of DON concentrations occurred within the upper 10 m of the water column (Fig. 4c).

DCCHO concentrations varied from 0.2 µmol L$^{-1}$ to 4.2 µmol L$^{-1}$ (Fig. 3c), with highest concentrations near the surface. C contained in DCCHO represented from 1 to max. 25 % of DOC in the studied depth range. Amino sugars were represented solely by GlcN, as GalN was below DL in most samples. Acidic sugars were mainly represented by uronic sugars, i.e. GluUA and GalUA (Table 1), while GlcA was detected only sporadically. Overall, amino sugars and acidic sugars comprised 0.04±0.03 µmol L$^{-1}$ and 0.02±0.02 µmol L$^{-1}$, contributing 6±3 % and 3±2 % to DCCHO, respectively. Thus, the major part of DCCHO was represented by neutral sugars (Table 1). DHAA concentrations varied from 0.075 µmol L$^{-1}$ to 1.39 µmol L$^{-1}$ (Fig. 3d). Like for DCCHO, the highest DHAA concentrations were found above the oxycline, where C contained in DHAA represented 2±1 % DOC (max. 4 %) and nitrogen (N) contained in DHAA represented 15±14 % DON. Lowest DHAA concentrations were mainly found below 80 m depth and equivalent to ~1 %DOC and 6-8 %DON (Table 1). The major part of DHAA was represented by α-amino acids. The concentrations of GABA, which is commonly used as a signature of microbial activity (Davis et al., 2009), was very low in all samples and represented generally <1% of DHAA. In summary, the concentrations of all the DOM compounds were highest above the oxycline and the mean concentration gradients of the DOM compounds were restricted to a shallower depth compared to the mean gradient of O$_2$ (Fig. 4).

## 3.2 Diapycnal fluxes and supply

As outlined in the previous section vertical gradients of O$_2$, DOC, DON and their constituents were observed at 30 to 80 m depth in the study area. In this section we combine these vertical gradients with turbulence measurements to estimate the associated diapycnal fluxes and supply i.e. the diapycnal flux divergences.

For O$_2$, the mean diapycnal flux ($\overline{\nabla\Phi}_{O2}$) exhibited a maximum of 50 mmol O$_2$ m$^{-2}$ day$^{-1}$ at ~20m depth. It decreased over depth and vanished at 80m depth due to lack of vertical concentration gradients. Onshore (<40 km) and offshore (>40 km) O$_2$ fluxes did not differ statistically. This likely was due to the fact that while vertical oxygen gradients were enhanced in the offshore region (Fig.4a), the turbulence and, thus, eddy diffusivities were elevated in the onshore region. The mean diapycnal supply O$_2$ ($\overline{\nabla\Phi}_{O2}$), ranged from 1.2 µmol kg$^{-1}$ day$^{-1}$ at 10-24 m depth to near zero at 80 m depth (Table 2). Again, onshore (<40 km) and offshore (>40km) the diapycnal O$_2$ supply was not statistically different.

In contrary, mean diapycnal fluxes of DOC ($\overline{\Phi}_{DOC}$) was limited to shallower depth. Near the surface, $\overline{\Phi}_{DOC}$ was 31 mmol C m$^{-2}$ day$^{-1}$ and vanished already at ~50 m depth (Table 2). The diapycnal supply of DOC ($\overline{\nabla\Phi}_{DOC}$) exhibited a maximum of 1.8 µmol C kg$^{-1}$day$^{-1}$ at 10-38 m depth (1.5 times larger than $\overline{\nabla\Phi}_{O2}$.) (Table 2, Eq. 4). Compared to NPP, estimated to 3.9 (0.6-8.6) gC m-2 day-1 for our study area and period, the DOC flux represented from a maximum of ~10 %NPP at ~20 m depth to near zero %NPP at ~50 m depth. As it was mentioned in the section 3.1, we did not find a vertical DON gradient, resulting in very low diapycnal DON fluxes and supply estimates (Table 2). However, N fluxes were obtained from DHAA

transport. Mean C and N fluxes via DCCHO and DHAA ranged from near zero below 30-40 m depth to 6 mmol C $m^{-2}$ $day^{-1}$ ($\overline{\Phi}_{DCCHO(C)}$), 0.9 mmol C $m^{-2}$ $day^{-1}$ ($\overline{\Phi}_{DHAA(C)}$ )and 0.3 mmol N $m^{-2}$ $day^{-1}$ ($\overline{\Phi}_{DHAA(N)}$) at 10-20m depth (Table 2). The diapycnal C and N supply via DCCHO and DHAA ranged from near zero to a maximum of 0.4 µmol C $kg^{-1}day^{-1}$ ($\overline{\nabla\Phi}_{DCCHO(C)}$), 0.06 µmol C $kg^{-1}day^{-1}$($\overline{\nabla\Phi}_{DHAA(C)}$), and 0.02 µmol N $kg^{-1}day^{-1}$($\overline{\nabla\Phi}_{DHAA(N)}$) at 10-38 m depth. The diapycnal

C supply via DCCHO and DHAA at its maximum comprised ~38% of $\overline{\nabla\Phi}_{O2}$, when estimated by Eq. (4). In summary, our diapycnal flux and supply calculation revealed that the diapycnal $O_2$ supply reaches deeper into the oxycline than the diapycnal DOM supply. This is especially true for DCCHO and DHAA, representing the labile and semi-labile parts of DOM.

### 3.3 Linking the DOM composition and the utilization of $O_2$

To understand whether low-$O_2$ conditions of the OMZ may cause changes in DOM composition, we complement our quantitative estimates of the DOM and $O_2$ supply with the analyses of DOM quality. For this, the composition of neutral DCCHO and DHAA via PCA was compared to environmental factors, i.e. temperature, AOU and salinity, and to organic matter composition from the well oxygenated water column as described in Kaiser and Benner (2009). The first principle component (Dim 1) (Fig. 5, "variables factor map") of the PCA was strongly influenced by AOU, indicating the interrelation

of the DOM composition and removal of $O_2$. The utilization of $O_2$ was accompanied by selective removal of Glu, Phe, Leu, ILeu and Ser, and Rha, Gal, and Fuc (Fig. 5, Table 1). Gly, Thr and Glc mol% were increasing along with increase in AOU (Fig. 5). In general, the composition of DOM from the surface samples from our study was similar to the composition of DOM from the samples, collected from well oxygenated open ocean sites by Kaiser and Benner (2009), as the individual scores of the samples cluster together on Dim.1 of the PCA (Fig. 5, "individuals factor map"). The samples, collected within

the OMZ were much poorer in composition, even in comparison to the deepest open ocean samples (~4000m), as they grouped from the negative side of Dim. 1.

The differences on the second dimension of PCA (Dim.2) were driven likely by regional differences in the DOM composition, i.e. by mol% of Ala, Arb, and Fuc, and distributions of mol% Asp, Phe, Val and Leu over depth (Fig. 5, Table 1, Kaiser and Benner, 2009).

**4 Discussion**

The observed distributions of $O_2$ and of DOC and DON components are the result of sinks and sources in the water column, mainly due to microbial processes and isopycnal and diapycnal supply (i.e. flux divergences) controlled by physical processes. A quantification of each of those individual processes is essential for understanding of important mechanisms, controlling $O_2$ and organic matter cycling off Peru and, therefore, the formation and maintenance of the Peruvian OMZ.

Previous studies have shown that turbulent mixing processes in the eastern boundary upwelling systems (EBUS) are strongly enhanced and that the resulting diapycnal supply is often a leading term in the flux divergence balances of $O_2$, nutrients and other solutes in the upper ocean (e.g. Schafstall et al., 2010; Kock et al., 2012; Brandt et al., 2015; Steinfeldt et al., 2015).

The diapycnal $O_2$ and DOM fluxes and supply determined in this study represent average values for the continental margin ranging from the shelf to about 100 km offshore. This spatial averaging is likely responsible for a lower near-surface diapycnal $O_2$ flux (50 mmol$O_2$ m$^{-2}$ day$^{-1}$) compared to other EBUS. For example, Brandt et al. (2015b) determined a near-surface diapycnal $O_2$ flux of 73 mmol$O_2$ m$^{-2}$ day$^{-1}$ in the Mauritanian upwelling during the high productivity season in boreal winter. In their study, the diapycnal $O_2$ flux was able to sustain benthic respiration on the continental shelf down to a bottom depth of 100 m. Herewith, the diapycnal $O_2$ supply, found in our study, was of similar magnitude as the rates of $O_2$ consumption (~1 µmol kg$^{-1}$ day$^{-1}$) determined by *in situ* incubations at 50-80 m water depth during the Austral summer season in the ETSP off Peru (Kalvelage et al., 2015) and similar estimates for North and South Pacific OMZs (Revsbech et al., 2009 and Tiano et al., 2014).

Other terms of the $O_2$ transport budget, such as isopycnal supply by meso- (Thomsen et al., 2016a) and submesoscale (Thomsen et al., 2016b) dynamics, or fluxes due to upwelling (e.g. Steinfeldt et al., 2015) might play an important role for the distribution of $O_2$ in the upper ocean, particularly in the region of the continental slope and the shelf. In turn, the deep chl *a* maximum, formed by photosynthetic cyanobacteria, i.e. *Prochlorococcus*, that have been found in the ETSP (Lavin et al, 2010; Ulloa et al., 2012; Meyer et al., 2017) may provide an additional $O_2$ source at depth. Furthermore, the presented diapycnal fluxes and supply of $O_2$ were determined from the data collected during ocean settings typical for the Austral summer season of non-El Niño/ La Niña-year. In the water column, $O_2$ concentrations and background settings for the production of turbulence were shown to vary substantially on seasonal and interannual time scales (e.g. Graco et al., 2017). Thus, the diapycnal fluxes and supply of $O_2$ shall vary on the same timescales. Therefore, our results should be considered as the first estimates of diapycnal $O_2$ fluxes and supply in the ETSP off Peru during Austral summer season during non-El Niño/ La Niña regime.

Like for $O_2$, the transport of DOM through the water column is achieved by advective and diffusive transport processes. Therefore, along with turbulent mixing, other transport terms will also take their part in shaping the DOM distribution off Peru. For instance, vertical advection (i.e. upwelling) transports deep water, which is characterized by highly altered DOM and low DOC concentrations, into the upper ocean near the continental margins. The upwelling may counteract the turbulent downward flux of DOC and, therefore, contribute to a "compression" or sharpening of the vertical DOM concentration and composition profiles. This is unique to upwelling systems and different to the open ocean regions where low DOC concentration gradients and smaller changes in the DOM composition were observed at similar depth (Kaiser and Benner, 2009). Additionally, meso- (Thomsen et al., 2016a) and submesoscale (Thomsen et al., 2016b) dynamics have been observed in the studied area. They were shown to modify nutrient and $O_2$ distributions by stirring the water across continental slope and likely influence the DOM distribution off Peru too. However, no quantitative information on DOM fluxes, associated with upwelling, meso- or submesoscale dynamics off Peru are available to date. Seasonal and interannual variations in

physical dynamics may as well affect DOM distribution off Peru, e.g. deepening of the mixed layer during Austral winter (Echevin et al., 2008) or intense downwelling/upwelling during El Niño/La Niña events (e.g. Graco et al., 2017) may result in the diapycnal DOM supply to a different depth than during typical Austral summer season.

DOM might also be transported to depth within particles. Thus, the "uncoupled" dissolution of large sinking aggregates as a
result of bacterial enzymatic activity (Smith et al., 1992) or abiotically (Sempéré et al., 2000) may serve as an additional DOM source and, therefore, affect the distribution of DOM in the water column. The sporadic dissolution of particles may bias the diapycnal DOM flux estimates at individual stations. Therefore, the bias may be reduced by calculating the mean diapycnal flux over a large number of depth profiles. The continuous DOM release from POM over the water column (e.g. Lefèvre et al., 1996), in turn, may lead to an overestimation of diapycnal DOM fluxes and DOM based microbial respiration.
However, no direct measurements of the DOM fraction resulting from particle dissolution exist so far in the studied area. Furthermore, DOM is affected by other abiotic or biological processes in the water column. For instance, the observed very low diapycnal DON flux may suggest a DON removal in the upper water column. Low concentrations of inorganic nutrients above 20 m depth (Thomsen et al., 2016a), and an overall nitrogen limitation that was found to be characteristic for the surface communities in the ETSP off Peru (Franz et al., 2012b), might force those communities to switch to organic nitrogen
sources (e.g. Bradley et al., 2010), therefore reducing DON in the upper water column. Photoreactions could also reduce DON incorporated into large chromophoric molecules through production of volatile N compounds or inorganic N (Zepp et al., 1998). Thus, DOM composition was suggested to be affected by the photochemistry in our study area (Galgani and Engel, 2016, Loginova et al., 2016). Photochemical degradation to CO, $CO_2$ and other volatile compounds (Zepp et al., 1998) could lower the near surface diapycnal DOC flux, as well.
Herewith, our data suggest that the diapycnal DOC flux in the upper 20m of the water column off Peru is in the same order of magnitude as the diapycnal $O_2$ flux (Table 2). The annual diapycnal DOC flux (2.7 mol C $m^{-2}yr^{-1}$) into the upper OMZ, estimated from our results by averaging $\overline{\Phi}_{DOC}$ above the mean depth of the oxycline (from below the mixed layer to 80 m depth) and integrating over a year, is in the same order of magnitude as previously reported data for the North Pacific Subtropical Gyre, where DOC export was estimated by a mass balance approach (1.6-2.7 molC $m^{-2}$ $yr^{-1}$; Emerson et al.,
1997) and by fitting an exponential decay function over depth (0.5±0.1 molC $m^{-2}$ $yr^{-1}$; Kaiser and Benner 2012). ). Compared to NPP, the diapycnal DOC flux (~10% NPP) was comparable to the POC export, previously reported for the upper water column in the ETSP off Chile (~12 %NPP (30 m depth), Pantoja et al., 2004), and in the ETSP off Peru (~6 %NPP (52 m depth); Gagosian et al., 1983; 16-42% NPP (near the surface); Kalvelage et al., 2013), advocating turbulent mixing of DOM to be an important C export mechanism in the upper oxycline.
Furthermore, in the upper water column (from below the mixed layer to 38 m water depth), the diapycnal DOC supply was higher, than the diapycnal $O_2$ supply, suggesting that DOC respiration could exhaust all $O_2$. However, the vanishing of DOC flux above the upper oxycline suggests that the bioavailable fraction of DOM is respired well before entering the upper OMZ. This is even more apparent, when considering diapycnal DHAA and DCCHO fluxes, which decayed more rapidly compared to the diapycnal DOC flux, suggesting preferential uptake of DHAA and DCCHO in the water column. The

diapycnal supply of DHAA and DCCHO could not fully explain the diapycnal supply of DOC, as those were responsible for only ~26% of $\overline{\nabla\Phi}_{DOC}$ when summed up together. This may hint to a presence of an additional bioavailable DOM component that was respired in the water column, and/or to other DOM removal mechanisms in the near-surface waters. For instance, DOM may form marine microgels and hence POM (Chin et al., 1998; Engel et al., 2004, Verdugo et al., 2004) or be trapped in the pore space of already existing particles (e.g. Benner, 2002).

As DHAA and DCCHO are preferentially utilized during microbial decomposition of organic matter (Skoog and Benner, 1997; Lee et al., 2000; Amon et al., 2001), their carbon yield (%DOC) and composition may serve as indicators of diagenetic history of DOM (e.g. Kaiser and Benner, 2009; Davis et al., 2009). Thus, the relatively high carbon yield of DHAA and DCCHO (Table 1), found near the surface during our study, suggests that DOM in surface waters off Peru is more bioavailable, compared to the open ocean (Davis and Benner, 2007; Kaiser and Benner, 2009). It is, however, rapidly altered at shallow depth. Applying the classification of Davis and Benner (2007), that implies that carbon yields of DHAA above 1.6 %DOC and 1.09 %DOC are corresponding to labile and semi-labile DOM, respectively, to our data suggests that the labile and semi-labile DOM off Peru was restricted upper 50 m of the water column.

The compositional analyses of DHAA and DCCHO suggested preferential microbial uptake of Glu, Phe, Ser, Leu and Rha, Gal, Fuc, Ara in the near surface waters, as below 50 m depth, the composition of DHAA and DCCHO were dominated by Gly and Glc, respectively (Fig. 5, Table 1). Glc was previously suggested to be less susceptible to microbial degradation compared to preferentially removed Fuc, Gal, and Ara (Ittekot et al., 1981; Sempere et al., 2008; Goldberg et al., 2010; Engel et al., 2012). Enrichment in Gly with depth has also been proposed to reflect the low nutritional value of Gly in anoxic sediments off Chile (Pantoja and Lee, 2003) and in sediments of the North Sea (Dauwe and Middelburg, 1998). Therewith, our data suggest that DOM in the shallow OMZ off Peru was characterized by stronger alteration compared to open ocean samples (Kaiser and Benner, 2009) at even much greater depths (up to 4000m). This may be due to both, an upwelling of altered DOM from the deep and a rapid and very extensive heterotrophic DOM utilization in the ETSP. The upwelling may "compress" labile and semi-labile DOM towards the surface, while the rapid microbial utilization of DOM shall prevent labile and semi-labile DOM export into the OMZ, and also would imply a pronounced heterotrophic respiration. The latter was suggested by our PCA analyses, as DOM composition was highly interrelated to AOU. Herewith, the diapycnal supply of DHAA and DCCHO could explain up to 38% of $\overline{\nabla\Phi}_{O2}$. This suggest, that despite the diapycnal fluxes of labile and semi-labile fractions of DOM may not reach deep into the core of the OMZ, DOM based microbial respiration above the OMZ may substantially attenuate the diapycnal $O_2$ flux that ventilates the upper oxycline. In other words, DOM may alter the shape of the upper oxycline, and, therefore, contribute to the formation and maintenance of the OMZ.

# 5 Conclusions

Our results suggest that DOM, i.e. DCCHO and DHAA, is significantly consumed and altered above the upper oxycline in the ETSP off Peru. Thus, despite the presence of high DOC concentrations in the euphotic zone, DOM may enter the OMZ

in an already highly reworked stage. Herewith, DOM respiration may contribute substantially (~38%) to $O_2$ reduction in the upper water column, potentially controlling the shape of the upper oxycline of the OMZ. The elevated diapycnal supply of DOC to the upper oxycline, which cannot be explained by microbial processes solely, hint to the presence of an additional DOM removal mechanism, such as microgel formation or absorption onto particles.

## 6 Data availability

The microstructure profiles are available at https:/doi.org/10.1594/PANGAEA.868400. The $O_2$, temperature, salinity, chl *a* fluorescence and nutrients were published at https:/doi.org/10.1594/PANGAEA.860727. The DOM data will be available at PANGAEA (www.pangaea.de, search project: sfb754) after publication.

## 7 Competing interests

The authors of this manuscript are not aware of any real or perceived financial conflicts of interests for other authors or authors that may be perceived as having a conflict of interest with respect to the results of this paper.

## 8 Acknowledgments

This study was supported by the Deutsche Forschungsgemeinschaft (dfg.de) trough SFB754 "Climate-Biogeochemical Interactions in the Tropical Ocean" (subproject B9) and CP1403 "Transfer and remineralization of biogenic elements in the tropical oxygen minimum zones".
We thank the chief scientists of the M93 cruise G. Lavik and T. Kanzow for station planning and support during sampling, as well as the crew and scientists onboard. We are also grateful to G. Krahmann for processing the CTD data, to C. Mages and R. Flerus for help with the water sampling, to J. Roa and R. Flerus for technician support. Special thanks to F. Le Moigne for help with the NPP calculation. We thank anonymous referee 1 and R. Benner for their valuable comments on improving the manuscript.

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

**Table 1: Relative composition (mol%) of dissolved hydrolysable amino acids (DHAA) and dissolved combined carbohydrates (DCCHO) in the water column, "n.d." - not detectable. Abbreviations "nS", "SN" and "SA" stand for neutral sugars, amino sugars and acidic sugars, respectively. The number of samples at each depth interval, used for calculation of the average value, is given as "n". The mean values for DHAA and DCCHO composition below the mixed layer (10 to 122 m) are reported for similar depth intervals (14 m) as diapycnal DOM and $O_2$ fluxes. The mean values for DHAA and DCHO within the mixed layer are reported for ~5 m depth intervals.**

| Depth (m) | n | DHAA | | | mol% DHAA | | | | | | | | | | | |
|---|---|---|---|---|---|---|---|---|---|---|---|---|---|---|---|---|
| | | (µmol L⁻¹) | (%DOC) | (%DON) | Gly | Thr | Ala | Asp | Glu | Ser | Arg | Leu | Val | Ileu | Phe | Tyr |
| 1-5 | 30 | 0.6±0.3 | 2±1 | 15±10 | 22±4 | 9±1 | 11±1 | 17±1 | 15±3 | 11±2 | 2.3±0.3 | 4±1 | 3.0±0.4 | 2.5±0.6 | 2.4±0.4 | 1.8±0.4 |
| 5-10 | 25 | 0.5±0.3 | 2.3±0.9 | 15±9 | 23±4 | 9±2 | 11±1 | 17±1 | 15±4 | 10±1 | 2.2±0.4 | 4±1 | 2.9±0.6 | 2.1±0.5 | 2.1±0.4 | 1.7±0.3 |
| 10-24 | 48 | 0.4±0.2 | 1.8±0.8 | 16±14 | 25±4 | 9±2 | 11±1 | 17±1 | 13±2 | 9±1 | 2.1±0.6 | 3±1 | 2.8±0.7 | 2.2±0.7 | 2.1±0.6 | 1.9±0.5 |
| 24-38 | 28 | 0.24±0.07 | 1.2±0.3 | 12±14 | 28±3 | 10±1 | 12±1 | 17±1 | 11±2 | 9±1 | 1.9±0.4 | 3±1 | 2.4±0.6 | 1.9±0.6 | 1.8±0.4 | 2.0±0.7 |
| 38-52 | 34 | 0.20±0.05 | 1.0±0.4 | 9±7 | 29±6 | 10±2 | 12±2 | 16±2 | 11±2 | 9±1 | 1.8±0.6 | 3±2 | 2.3±0.8 | 1.7±0.5 | 1.8±0.4 | 1.7±0.4 |
| 52-66 | 35 | 0.17±0.03 | 0.9±0.3 | 13±19 | 31±3 | 10±2 | 12±1 | 16±1 | 10±2 | 8±1 | 1.7±0.4 | 2±1 | 2.4±0.5 | 1.6±0.7 | 1.7±0.3 | 1.7±0.5 |
| 66-80 | 27 | 0.16±0.05 | 0.9±0.3 | 9±8 | 32±4 | 10±1 | 12±1 | 15±2 | 10±2 | 8±1 | 1.7±0.5 | 2±1 | 2.5±0.6 | 1.7±0.8 | 1.7±0.4 | 1.8±0.5 |
| 80-94 | 22 | 0.15±0.08 | 0.9±0.4 | 8±7 | 34±3 | 10±2 | 12±2 | 15±1 | 10±2 | 9±1 | 1.6±0.4 | 2±1 | 2.2±0.7 | 1.3±0.7 | 1.6±0.4 | 1.6±0.4 |
| 94-108 | 14 | 0.13±0.03 | 0.7±0.2 | 9±8 | 34±3 | 10±2 | 13±2 | 15±2 | 9±2 | 8±2 | 1.6±0.5 | 2±1 | 2.3±0.7 | 2±1 | 1.7±0.4 | 1.7±0.9 |
| 108-122 | 13 | 0.13±0.03 | 0.8±0.2 | 6±4 | 32±3 | 10±2 | 12±1 | 16±2 | 10±2 | 8±1 | 1.7±0.3 | 3±1 | 2.3±0.8 | 2±1 | 1.9±0.4 | 1.7±0.5 |
| 122-200 | 18 | 0.12±0.03 | 0.7±0.3 | 8±6 | 35±3 | 10±1 | 12±2 | 15±2 | 9±1 | 8±2 | 1.5±0.7 | 2±1 | 2.5±0.6 | 1.7±0.7 | 1.5±0.4 | 1.5±0.5 |

| Depth (m) | n | DCCHO (µmol L⁻¹) | | | mol%DOC | | | mol% nS | | | | | | mol% SA | | |
|---|---|---|---|---|---|---|---|---|---|---|---|---|---|---|---|---|
| | | nS | SN | SA | nS | SN | SA | Glc | ManXyl | Gal | Rhm | Fuc | Ara | GluUA | GalUA | GlcA |
| 1-5 | 30 | 1.5±0.8 | 0.10±0.03 | 0.10±0.08 | 9±4 | 0.6±0.2 | 0.6±0.3 | 30±13 | 32±6 | 17±6 | 11±8 | 8±2 | 2±1 | 48±21 | 51±21 | 0.4±2 |
| 5-10 | 25 | 1.1±0.6 | 0.08±0.03 | 0.07±0.05 | 8±4 | 0.5±0.1 | 0.5±0.3 | 33±11 | 33±5 | 16±6 | 8±6 | 8±2 | 2±1 | 43±26 | 55±24 | 2±10 |
| 10-24 | 47 | 0.7±0.3 | 0.06±0.02 | 0.04±0.03 | 5±2 | 0.4±0.1 | 0.3±0.2 | 36±13 | 37±8 | 12±5 | 5±4 | 7±2 | 2±1 | 32±25 | 67±25 | 1±7 |
| 24-38 | 28 | 0.4±0.1 | 0.04±0.01 | 0.02±0.02 | 4±1 | 0.3±0.1 | 0.2±0.1 | 43±11 | 38±7 | 9±4 | 2±2 | 6±2 | 0.4±1.0 | 20±20 | 80±20 | n.d. |
| 38-52 | 35 | 0.4±0.2 | 0.03±0.01 | 0.02±0.01 | 4±2 | 0.3±0.1 | 0.1±0.1 | 42±10 | 41±9 | 9±3 | 2±2 | 5±2 | 0.3±0.8 | 28±30 | 72±30 | n.d. |
| 52-66 | 34 | 0.5±0.2 | 0.03±0.01 | 0.02±0.02 | 4±2 | 0.2±0.1 | 0.2±0.2 | 45±9 | 41±9 | 7±4 | 2±2 | 5±2 | 0.2±0.6 | 21±27 | 77±28 | 2±11 |
| 66-80 | 27 | 0.4±0.2 | 0.02±0.01 | 0.01±0.01 | 4±2 | 0.2±0.1 | 0.1±0.1 | 47±13 | 44±12 | 5±3 | 1±1 | 3±2 | 0.3±0.7 | 19±28 | 81±28 | n.d. |
| 80-94 | 22 | 0.4±0.2 | 0.02±0.01 | 0.01±0.01 | 4±2 | 0.2±0.1 | 0.1±0.1 | 47±11 | 45±10 | 4±3 | 0.1±0.6 | 2±2 | 0.7±1.3 | 32±33 | 68±33 | n.d. |
| 94-108 | 15 | 0.3±0.1 | 0.02±0.01 | 0.01±0.01 | 3±1 | 0.2±0.1 | 0.1±0.1 | 53±11 | 40±10 | 4±3 | 0.1±0.5 | 2±2 | 0.2±0.9 | 28±29 | 72±29 | n.d. |
| 108-122 | 13 | 0.4±0.1 | 0.02±0.01 | 0.02±0.02 | 4±2 | 0.2±0.1 | 0.2±0.2 | 51±16 | 43±14 | 3±3 | 0.2±0.7 | 2±2 | 0.3±1.0 | 44±46 | 56±46 | n.d. |
| 122-200 | 18 | 0.4±0.2 | 0.02±0.01 | 0.01±0.02 | 4±1 | 0.2±0.1 | 0.1±0.2 | 52±10 | 44±9 | 2±2 | n.d. | 1±2 | 0.7±2.3 | 22±30 | 78±30 | n.d. |

**Table 2: Diapycnal fluxes and supplies (in bold) of $O_2$ and DOM: DOC, DON, dissolved organic carbon in DCCHO and DHAA and dissolved organic nitrogen in DHAA. 95% confidence intervals, calculated after Schafstall et al. (2010) for each parameter, are presented in brackets. BLM – "below the mixed layer" – a depth, defined below 10m of the water column, using a threshold criterion of 0.2°C temperature decrease.**

| | Depth (m) | DOC | DON | DCCHO-C | DHAA-C | DHAA-N | $O_2$ |
|---|---|---|---|---|---|---|---|
| **Flux (mmol $m^{-2}$ $day^{-1}$)** | BML-24 | 31 (+56/-6) | -0.6 (+0.1/-1.0) | 6 (+8/-0.06) | 0.9 (+1.3/+0.1) | 0.3 (+0.4/+0.05) | 50 (+77/+17) |
| | 24-38 | 5 (+24/-12) | 8 (+87/-2) | 0.2 (+6/-0.01) | 0.07 (+0.4/+0.03) | 0.03 (+0.15/+0.013) | 32 (+77/+11) |
| | 38-52 | 0.4 (+1.2/-0.1) | 0.4 (+8/-1) | 0.12 (+2/+0.04) | 0.07 (+0.3/+0.04) | 0.03 (+0.1/+0.01) | 32 (+72/+15) |
| | 52-66 | 0.2 (+0.6/-0.003) | 0.5 (+14/-2) | 0.01 (+16/-0.9) | 0.05 (+0.2/+0.03) | 0.02 (+0.1/+0.01) | 17 (+89/+5) |
| | 66-80 | 0.6 (+1.8/-0.03) | 0.1 (+12/-2) | 0.12 (+11/-0.5) | 0.02 (+0.5/-0.08) | $0.7\times10^{-2}$ (+0.2/-0.03) | 8 (+17/+1) |
| | 80-94 | -0.5 (+0.3/-0.4) | $-0.1\times10^{-2}$ (+0.01/-0.06) | 0.14 (+11/-0.5) | 0.01 (+0.2/-0.02) | $0.4\times10^{-2}$ (+0.06/-0.01) | 0.12 (+0.2/+0.03) |
| | 94-108 | -0.2 (+0.02/-0.4) | 0.05 (+11/-2) | 0.09 (+24/-1) | $0.6\times10^{-2}$ (+0.3/-0.05) | $0.2\times10^{-2}$ (+0.1/-0.02) | 0.016 (+0.04/+0.01) |
| | 108-122 | -0.2 (-0.06/-0.4) | 0.01 (+3/-0.5) | -0.01 (+0.3/-4) | $0.2\times10^{-3}$ (+0.01/-0.02) | $0.1\times10^{-3}$ (+0.01/-0.001) | 0.02 (+0.06/+0.01) |
| **Supply (µmol $kg^{-1}$ $day^{-1}$)** | BML-38 | 1.8 (+4.0/-1.0) | -0.6 (+5/-1) | 0.4 (+0.8/-0.02) | 0.06 (+0.09/+0.005) | 0.02 (+0.03/+0.002) | 1.2 (+5/-2) |
| | 24-52 | 0.3 (+1.6/-0.9) | 0.6 (+6/-0.2) | $0.5\times10^{-2}$ (+0.4/-0.01) | $0.1\times10^{-3}$ (+0.02/-0.003) | $0.2\times10^{-4}$ (+0.01/-0.001) | 0.04 (+4/-2) |
| | 38-66 | 0.01 (+0.07/-0.03) | -0.01 (+1/-0.2) | $0.8\times10^{-2}$ (+0.2/+0.002) | $0.1\times10^{-2}$ (+0.02/-0.001) | $0.5\times10^{-3}$ (+0.01/$-6\times10^{-4}$) | 1.0 (+7/-0.5) |
| | 52-80 | -0.03 (+0.05/-0.08) | 0.03 (+1/-0.2) | $-0.8\times10^{-2}$ (+1/-0.1) | $0.2\times10^{-2}$ (+0.04/-0.005) | $0.7\times10^{-3}$ (+0.01/-0.002) | 0.7 (+6/-0.3) |
| | 66-94 | 0.05 (+0.13/-0.006) | 0.01 (+0.9/-0.1) | $-0.1\times10^{-2}$ (+1/-0.1) | $0.6\times10^{-3}$ (+0.04/-0.007) | $0.2\times10^{-3}$ (+0.01/-0.003) | 0.5 (+1/+0.07) |
| | 80-108 | $0.8\times10^{-2}$ (+0.03/-0.02) | $-0.3\times10^{-2}$ (+0.8/-0.1) | $0.4\times10^{-2}$ (+2/-0.1) | $0.4\times10^{-3}$ (+0.02/-0.04) | $0.1\times10^{-3}$ (+0.007/-0.001) | $0.7\times10^{-2}$ (+0.01/+0.001) |
| | 94-122 | $0.4\times10^{-2}$ (+0.02/-0.01) | $0.2\times10^{-2}$ (+0.8/-0.1) | $0.7\times10^{-2}$ (+2/-0.3) | $0.4\times10^{-3}$ (+0.02/-0.004) | $0.1\times10^{-3}$ (+0.006/-0.001) | $-0.1\times10^{-2}$ (+0.003/-0.002) |

**Figure 1:** Study area and station map. CTD stations, where CTD-probe and fluorimeter measurements were accomplished are marked as black dots (a,b). PUMP-CTD stations are depicted in pink diamonds (a). CTD and PUMP-CTD stations, where DOM sampling was performed are marked as green stars (a). Microstructure measurements, combined with oxygen profiles are marked as grey circles (b). Microstructure measurements, combined with dissolved organic matter (dissolved organic carbon (DOC), dissolved hydrolysable amino acids (DHAA) and dissolved combined carbohydrates (DCCHO)) measurements marked as green pentagrams (b). Extra microstructure measurements, combined with DOC measurements marked with violet pentagrams (b). Shaded colors represent chl *a* concentrations at upper 10 m depth (a) and oxygen concentrations at 15m depth (b). Spaces between data points were interpolated by using TriScatteredInterp function (MATLAB, MathWorks).

**Figure 2:** Mean vertical distribution of the temperature (a), salinity (b), (c) chlorophyll *a* (chl *a*) and (d) $O_2$. $O_2$ values below 1 µmol $kg^{-1}$ are shaded in violet. The data from all transects and stations were averaged over intervals of 10 km on "Distance from the coast" axis and over 1 m on "Depth" axis. Isolines represent potential density, averaged over intervals of 10 km on "Distance from the coast" axis and over 1 m on "Depth" axis.

**Figure 3:** Dissolved organic carbon (DOC) (a), dissolved organic nitrogen (DON) (b), dissolved combined carbohydrates (DCCHO) (c) and dissolved hydrolysable amino acids (DHAA) (d) distributions over the water column. Data from all transects and stations were plotted against distance to coast (km). Space between data points was interpolated by using TriScatteredInterp function (MATLAB, MathWorks). Isolines represent potential density, averaged over intervals of 10 km on "Distance from the coast" axis and over 1 m on "Depth" axis.

**Figure 4:** Vertical distribution of $O_2$ (a), DOC (b), DON (c), DCCHO(C) (d), DHAA(C) (e), DHAA(N) (f). Black line and error bar represent mean distribution and standard deviations of the data points (grey circles), respectively. The blue and red lines and shaded areas represent the mean distributions and standard deviations of parameters onshore (<40 km) and offshore (>40 km), respectively.

**Figure 5:** The PCA analysis output: variables (on the left) and individuals scores of samples (from the right). The samples, collected above 50m depth are marked with acronym "s", the ones, below 50m depth – with acronym "d". The samples, which are used for comparison are marked with acronyms "HOT" and "BATS", and represented well oxygenated samples, collected from open Pacific and open Atlantic Oceans, respectively (Kaiser and Benner, 2009).

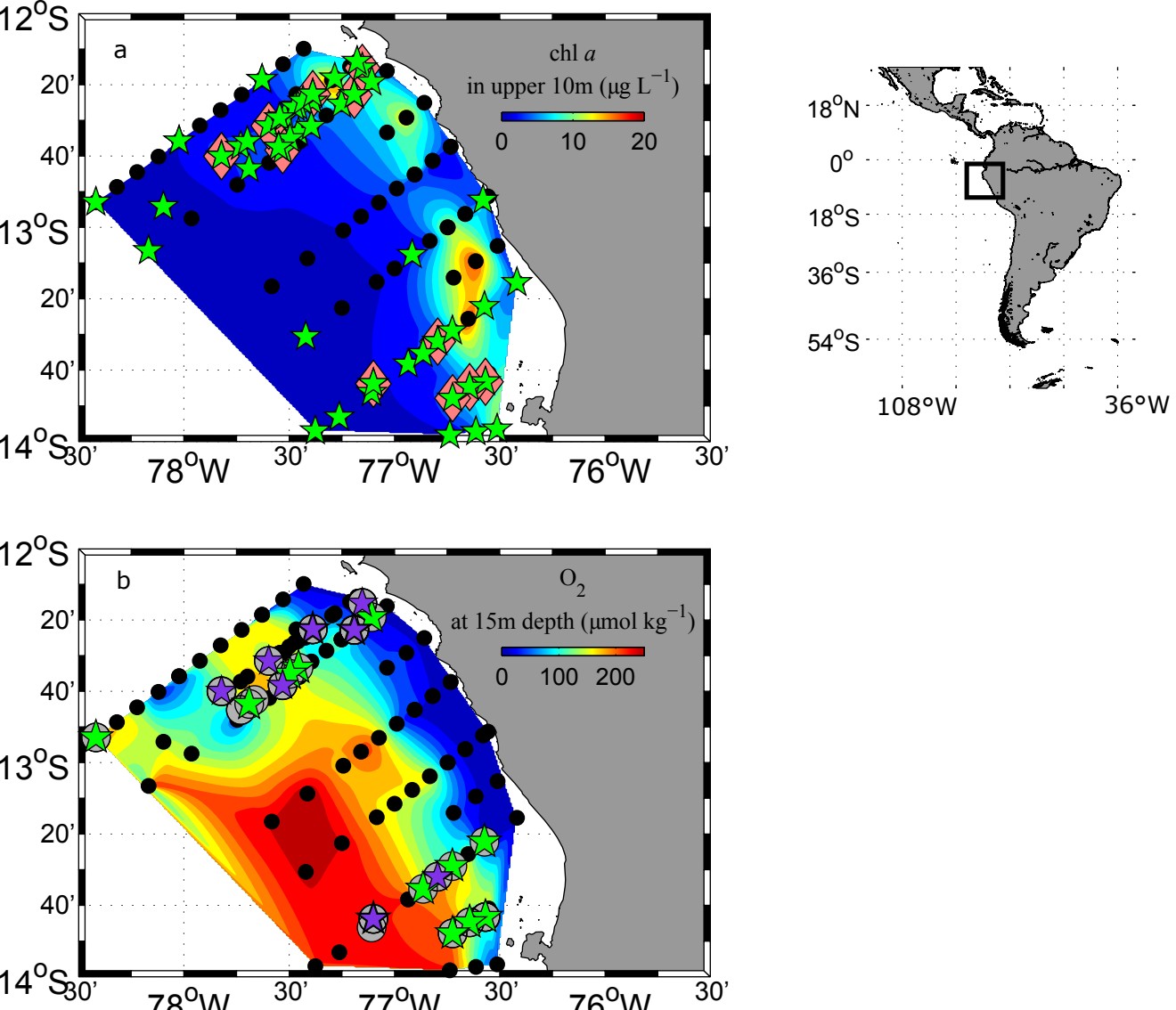

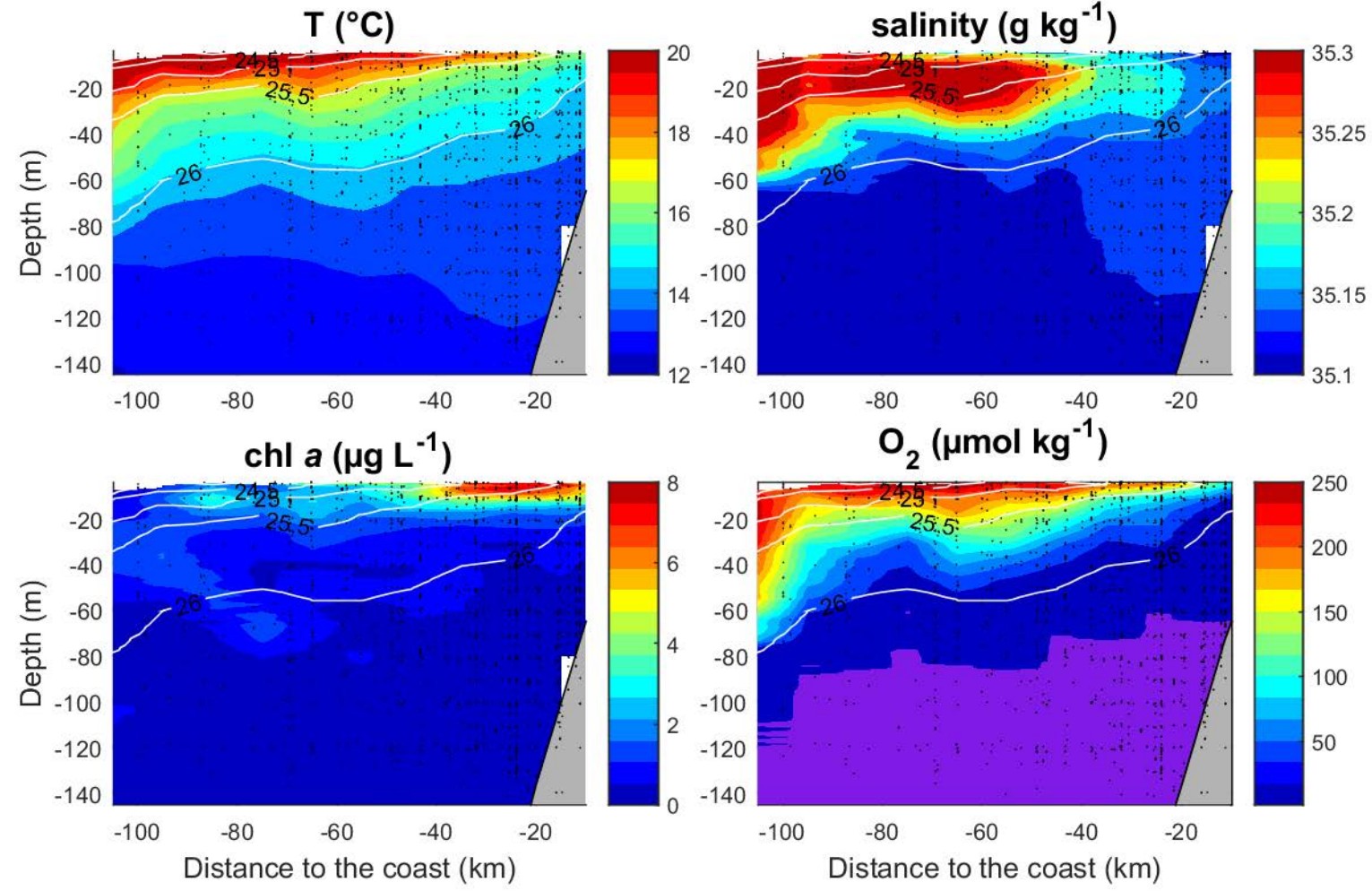

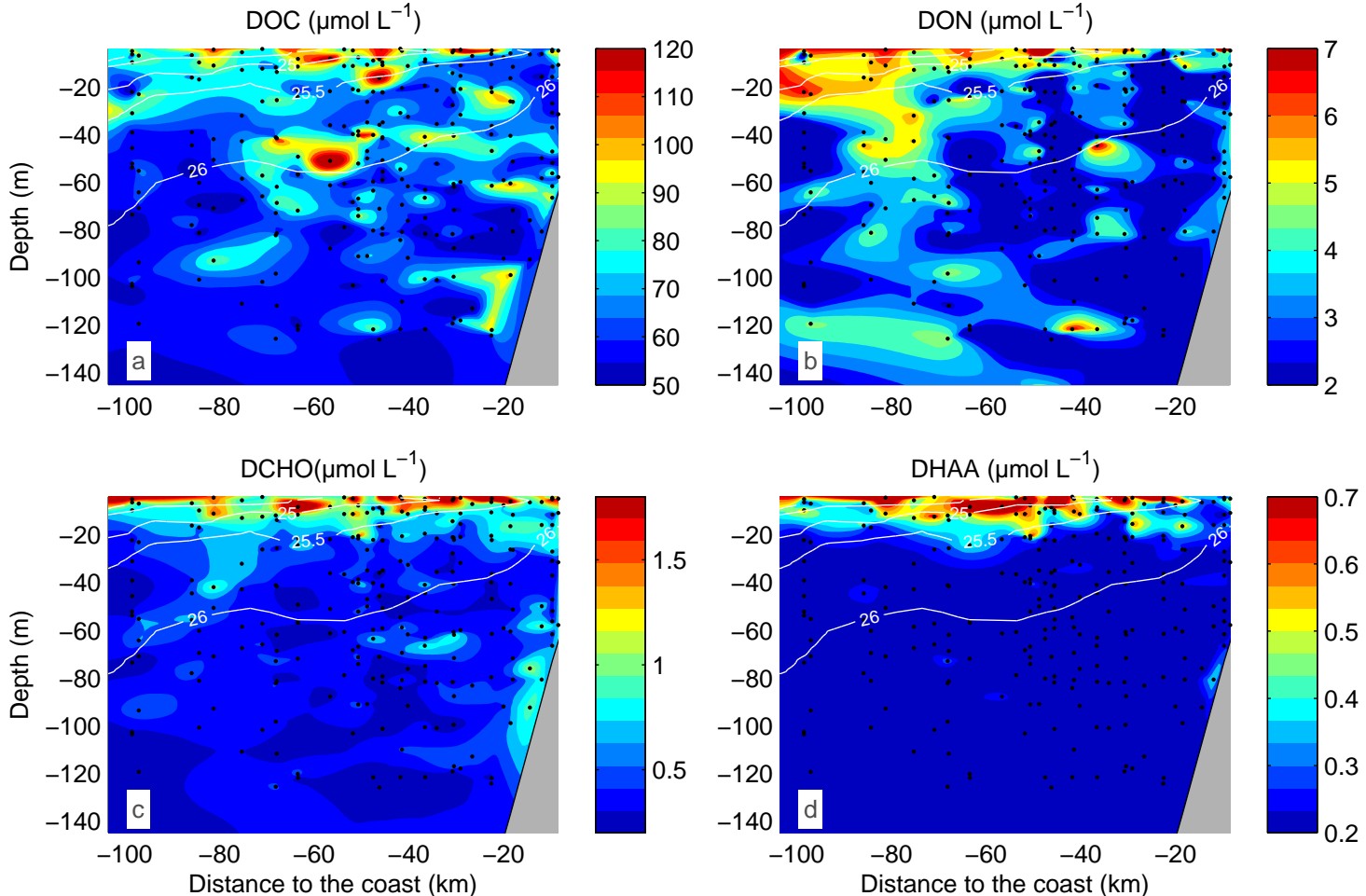

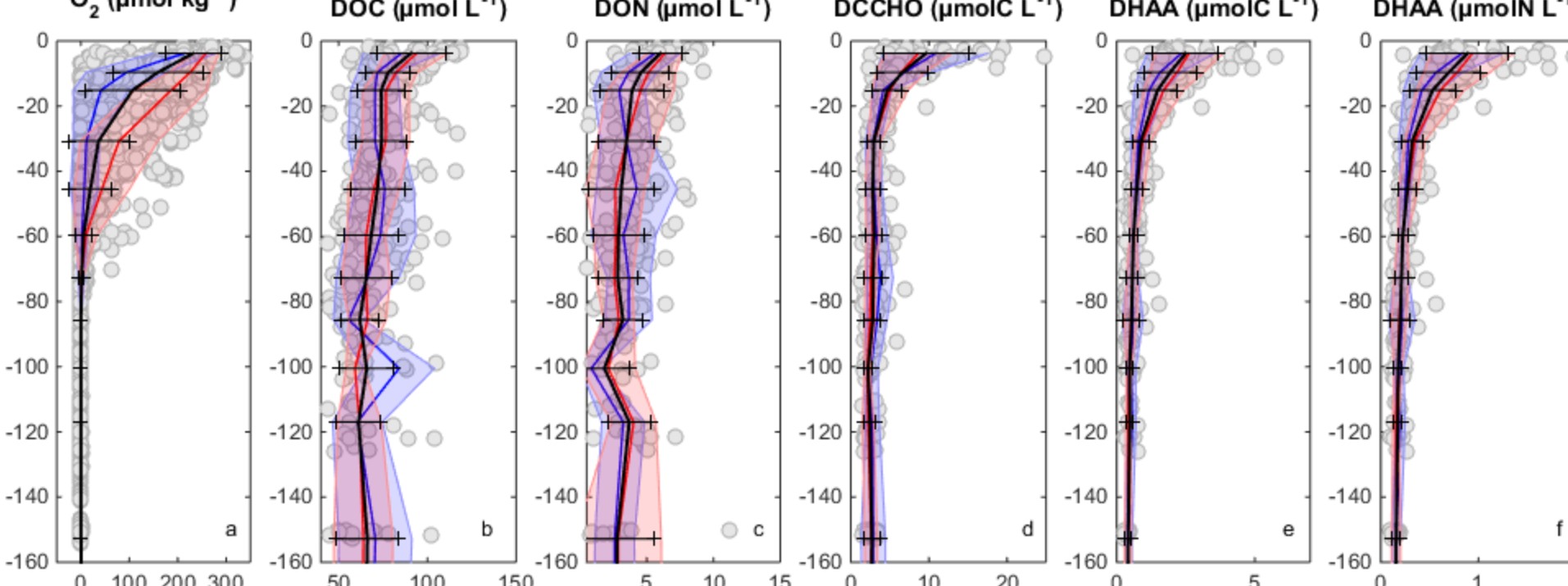

## Variables factor map (PCA)

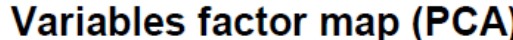

## Individuals factor map (PCA)