# Peer review of "Diapycnal dissolved organic matter supply into the upper Peruvian oxycline"

_Biogeosciences, 2018_

## Referee Comment (RC1) · Anonymous Referee #1 · 2 Jul 2018

General Comments:

The basic question posed in the Abstract was "how important is DOM utilization for O2 respiration within the Peruvian OMZ". The answer was not given unambiguously in the Abstract. The answer the authors should give in the Abstract, based on their results, is that "DOM introduced by vertical mixing has no role in contributing to O2 consumption in the core of the OMZ". This answer is given in the Discussion, but it is not in the Abstract. Instead, the authors state that "DOM utilization may play a significant role for shape of the upper Peruvian oxycline"; but that statement is not the answer to the question posed. The Abstract needs to be written for absolute clarity in terms of question and answer.

I did not find the outcomes of this work to be enlightening. We could see in the data

plots that DOC was high at the surface but low at 40, so clearly it was not surviving export by mixing to even 40 m depth. So its small (or non-existent) contribution to export into the OMZ core is pretty obvious just by looking at the distributions; the great effort by the authors to calculate vertical fluxes may have been excessive given the obvious answer to the question.

I'm not sure what is the main point of this paper. DOM is essentially not exported to the OMZ, but we did not need to see all the work done by the authors to know that outcome. That it contributes to the "shape of the upper oxycline" is the final finding given in the Abstract, but does that matter? The shape of the oxycline is not discussed elsewhere in the paper.

Specific Comments (Pg#/Line#)

1/31 "is one of the largest regions" In what regard? For an OMZ? And, "where the role of O2 concentrations discriminates." Discriminates what? And does an O2 concentration really have "a role"?

2/7 "anoxia-related processes" not enough information in that phrase.

2/26: "Accessing" should be "Assessing"

3/18 The acronym "GO" should be spelled out; presumably it is "General Oceanics"

6/2 What exactly is the "diapycnal solute supply"? This term should be explained fully, as it is central to the findings in the manuscript. Telling the reader that it is a 'divergence in flux' is inadequate.

6/25-26 Surface DOC concentrations >100 umol/L are not found in the ocean unless a river is nearby, which can add terrigenous DOC. The high values seem unrealistic. The values in the surface layer that are closer to 70 uM are more realistic, based on the data reported by Letscher et al. 2015 at nearby locations. The elevated DOC values at greater depth are suspect as well.

---

## Referee Comment (RC2) · R. Benner (Referee) · 8 Jul 2018

The authors present compelling physical and biogeochemical data indicating microbial utilization of DOM plays an important role in shaping the upper oxycline in the Peruvian upwelling system. Diapycnal fluxes of O2 and DOM from productive surface waters are estimated, and analyses of DOM concentrations and compositions indicate the microbial utilization of bioavailable components (e.g. amino acids and carbohydrates) occurs mostly in the upper 50 m of the water column.

In addition to the mol% compositional data presented for carbohydrates and amino acids, the DOC-normalized yields of neutral sugars and amino acids can provide insights about the bioavailability of DOC. These data should be presented in a table (e.g.

Table 1) or figure (e.g. Fig. 4). It appears carbohydrate and amino acid yields (%DOC) decline rapidly in the upper 50 m of the water column, indicating the preferential utilization of these bioavailable DOM components. The yields and bioavailability of DOC at 100 m can be compared to those at HOT and BATS to provide a more definitive indicator of the relative bioavailability and diagenetic state of DOM at these sites.

Observations of the low bioavailability and highly altered chemical composition of DOM at relatively shallow depths (<120 m) is likely due to upwelling of aged and altered DOM as well as active microbial utilization in surface waters (e.g. Steinfeldt et al. 2015). It appears upwelling compresses the vertical profiles of DOM concentration and composition. The manuscript would benefit from a discussion of the role of upwelling in shaping the observed biogeochemical distributions.

Specific comment:

The reported concentrations of the amino sugar, GalN, are very low in comparison to values in the north Pacific (HOT). The resulting GlcN:GalN ratios are extremely high (40-70). It appears there is a problem with the GalN measurements.

---

## Author Comment (AC1) · 10 Aug 2018

AC1: First of all, we are very thankful for Anonymous Referee #1 (AR1) for taking time for reading our manuscript and giving his/her comments for our study. In the following, comments of AR1 will be addressed one by one.

RC1: "The basic question posed in the Abstract was "how important is DOM utilization for O2 respiration within the Peruvian OMZ". The answer was not given unambiguously in the Abstract. The answer the authors should give in the Abstract, based on their results, is that "DOM introduced by vertical mixing has no role in contributing to O2 consumption in the core of the OMZ". This answer is given in the Discussion, but it is not in the Abstract. Instead, the authors state that "DOM utilization may play a

significant role for shape of the upper Peruvian oxcline"; but that statement is not the answer to the question posed. The Abstract needs to be written for absolute clarity in terms of question and answer."

AC1: We agree with AR1 that the abstract needs to be revised for clarification of our results: Abstract 9-10 The sentence: "However, the importance of DOM utilization for O2 respiration within the Peruvian OMZ remains unclear so far." will be changed to: "However, the importance of DOM utilization for O2 respiration in the Peruvian upwelling system in general and for shaping the upper oxcline in particular remains unclear so far." Abstract 10-16 The sentence: "Here, we evaluate the diapycnal fluxes of O2, dissolved organic carbon (DOC), dissolved organic nitrogen, dissolved hydrolysable amino acids (DHAA) and dissolved combined carbohydrates (DCCHO) and the composition of DOM in the ETSP off Peru to learn, whether labile DOM is reaching into the core of the OMZ and how important DOM utilization might be for O2 attenuation." will be changed to: "This study reports the first estimates of diapycnal fluxes and supply of O2, dissolved organic carbon (DOC), dissolved organic nitrogen, dissolved hydrolysable amino acids (DHAA) and dissolved combined carbohydrates (DCCHO) to the OMZ for the ETSP off Peru. Diapycnal flux and supply estimates were obtained by combining measured vertical diffusivities and solute concentration gradients. They were analysed together with the molecular composition of DCCHO and DHAA to infer the transport of labile DOM into the upper OMZ and the potential role of DOM utilization for the attenuation of the diapycnal O2 flux that ventilates the OMZ.". Abstract 19 The line: "suggesting that the labile DOM is already utilized" will be changed to: "suggesting that the labile DOM is extensively consumed" Abstract 24-25 The line: "which suggests that DOM utilization may play a significant role for shape of the upper Peruvian oxcline." will be replaced with: "which suggests that DOM utilization plays a significant role for shaping of the upper Peruvian oxcline" We feel that a modification of last sentence in Introduction (3/14) will help also for clarification of our results: The sentence: "Additionally, we analyze the composition of dissolved combined CHO and AA to learn, whether DOM and its labile and semi-labile constituents may be supplied to the core of the OMZ." will

be edited to: "Additionally, we analyze the composition of dissolved combined CHO (DCCHO) and dissolved hydrolysable AA (DHAA) to learn, whether DOM and its labile and semi-labile constituents may be supplied to the upper OMZ and the potential contribution of DOM based respiration to O2 flux attenuation."

RC1: I did not find the outcomes of this work to be enlightening. We could see in the data plots that DOC was high at the surface but low at 40, so clearly it was not surviving export by mixing to even 40 m depth. So its small (or non-existent) contribution to export into the OMZ core is pretty obvious just by looking at the distributions; the great effort by the authors to calculate vertical fluxes may have been excessive given the obvious answer to the question. I'm not sure what is the main point of this paper. DOM is essentially not exported to the OMZ, but we did not need to see all the work done by the authors to know that outcome. That it contributes to the "shape of the upper oxycline" is the final finding given in the Abstract, but does that matter? The shape of the oxycline is not discussed elsewhere in the paper.

AC1: We thank AR1 for taking the time to read our work and giving his/her opinion on our outcomes. We take this critical comment that our work is not "enlightening" as a motivation to make the importance of this first quantitative study on oxygen and DOM dynamics clearer for the reviewer and also for the reader. We are convinced that an accurate quantification of O2 and DOM fluxes and even more, the flux divergences (which are more informative for learning about sources and sinks of solutes) is an important contribution to the understanding of biogeochemical and microbial processes in OMZs and that the effort to calculate these fluxes should be valued. In particular, we do not share the statement of AR1 that "looking at the distributions" is sufficient to understand the complex O2 and organic matter dynamics off Peru. The distribution of a component, determined at one time-point does not provide any information on its processing and cannot give any quantitative information on matter fluxes. The ventilation of the OMZ by the physical supply of O2 from the surface ocean is constrained by the biological utilization of oxygen for respiration of OM that happens during transport process. It has been shown that aerobic and microaerobic microbial respiration is the main pathway of organic matter remineralization (Kalvelage et al., 2015) in the oxycline. In order to estimate the attenuation of O2 fluxes by microbial respiration of organic matter and by this the contribution of microbial processes to the formation and maintenance of the OMZ, we need to know the fluxes of O2 and the flux attenuation (supply), i.e. O2 uptake. Here, we give for the first time a quantitative estimate of those parameters and relate them (quantitatively!) to DOM supply (uptake) and to previously estimated oxygen consumption off Peru (Kalvelage et al., 2015). Please, note that the oxycline is considered as being part of the OMZ.

RC1: "DOM introduced by vertical mixing has no role in contributing to O2 consumption in the core of the OMZ ... DOM contributes to the "shape of the upper oxycline", but does that matter?".

AC1: As there is no oxygen to be consumed within the core of the OMZ, organic matter, if supplied to the core of the OMZ, cannot cause oxygen consumption, unless oxygen is supplied with it. Therefore, looking at O2 and DOM fluxes and their divergences is so important. Turbulent mixing is a major process for ventilating the OMZs, as ventilation by ocean currents is sluggish, and currents are carrying only low-O2 waters to the eastern boundary OMZs. Please see 9/10:" diapycnal supply is often a leading term in the flux divergence balances of O2, nutrients and other solutes in the upper ocean (e.g. Schafstall et al., 2010; Kock et al., 2012; Brandt et al., 2015; Steinfeldt et al., 2015)." Our findings suggest that a substantial part of O2 flux above the OMZ core is attenuated due to respiration of labile DOC. A process we refer to as "shaping of the oxycline" here. We will clarify our description of the O2 concentrations in the OMZ core in the revised manuscript: The sentence (4/6 Methods): "The O2 optode was calibrated by a combination of Winkler titration (Winkler, 1888; Hansen, 1999) and STOX sensor measurements (Revsbech et al., 2009)." will be changed to: "The O2 optode was calibrated by Winkler titration above the oxycline (Winkler, 1888; Hansen, 1999). The STOX sensor measurements, which revealed O2 concentrations of 0.01-0.05 $\mu$mol kg-

1 within the OMZ (Revsbech et al., 2009; Thomsen et al., 2016a) were used for O2 optode calibration at low O2 levels."

RC1: "DOM clearly not surviving export by mixing to even 40 m depth".

AC1: Our data represent mean values for the study area at the time of the field campaign and do not resolve episodic processes, which may occur e.g. through sub-mesoscale mixing and supply fresh DOM into the oxycline locally (Ulloa et al., 2012, Thomsen et al., 2016b). Moreover, deepening of the mixed layer and weakening of the stratification, as for instance in Austral winter, may potentially enhance DOM supply to the deeper waters (Thomsen et al., 2016b).

RC1: 1/31 "is one of the largest regions" In what regard? For an OMZ? And, "where the role of O2 concentrations discriminates." Discriminates what? And does an O2 concentration really have "a role"?

AC1: The sentence: "Due to the presence of a pronounced oxygen minimum zone (OMZ) (Karstensen et al., 2008), the eastern tropical South Pacific (ETSP) is one of the largest regions, where the role of O2 concentrations discriminates." will be changed to: "The eastern tropical South Pacific (ETSP) embodies one of the largest oxygen minimum zones (OMZ) in the world ocean (Karstensen et al., 2008; Paulmier and Ruiz-Pino, 2009)."

RC1: 2/7 "anoxia-related processes" not enough information in that phrase.

AC1: The phrase: "anoxia-related processes (Kalvelage et al., 2013)" will be changed to: "anoxia-induced processes, such as denitrification and anammox (Kalvelage et al., 2011, 2013)"

RC1: 2/26: "Accessing" should be "Assessing" AC1: "Accessing" will be replaced with "Assessing"

RC1: 3/18 The acronym "GO" should be spelled out; presumably it is "General Oceanics"

AC1: The line: "Seawater was sampled with a GO rosette" will be replaced with: "Seawater was sampled with a rosette (GO; General Oceanics, USA)"

RC1: 6/2 What exactly is the "diapycnal solute supply"? This term should be explained fully, as it is central to the findings in the manuscript. Telling the reader that it is a 'divergence in flux' is inadequate.

AC1: The diapycnal fluxes and supplies for all the dissolved parameters (solutes) were calculated by similar approach, therefore, the word "solutes" was used in the method for describing calculations used for dissolved oxygen, DOC, DHAA and DCCHO. At the interfaces sediment-water column/air-sea interface it makes sense to speak about fluxes to quantify exchange between reservoirs. Within the water column the change of fluxes over depth (or distance), i.e. the vertical flux attenuation referred to as flux divergences, indicates the rate of consumption or production and is the value that can be compared to sources and sinks. The flux divergence was described and calculated by equation 3. This value is an estimate for the diapycnal solute supply (an equivalent of solute consumption/remineralization), assuming that other sources or sinks (such as mesoscale, submesoscale or upwelling fluxes, or photochemical reactions) were negligible. We discuss, the potential influence of other sources and sinks (page9, line 20), however, these were not quantified in this study nor previously: "Other mixing terms of the O2 transport budget, such as isopycnal O2 supply by meso- (Thomsen et al., 2016a) and submesoscale (Thomsen et al., 2016b) dynamics or O2 fluxes due to upwelling (e.g. Steinfeldt et al., 2015) may provide an additional loss of O2 to the upper ocean, particularly in the region of the continental slope and the shelf. Furthermore, seasonal variations of the diapycnal solute fluxes may occur due to, for instance, deepening the mixed layer during winter season (Echevin et al., 2008). Therefore, our results should be considered as the first estimates of diapycnal fluxes and supply in ETSP off Peru during austral summer. Therefore, more observations shall improve the robustness of the flux estimates." and 9/29 "DOM transport through the water column is also restricted to advective and diffusive mixing processes. However, DOM is affected

also by other abiotic or biological processes in the water column. ... Photoreactions could also reduce DON incorporated into large chromophoric molecules through pro- duction of volatile N compounds or inorganic N (Zepp et al., 1998). . . . Photochemical degradation to CO, CO2 and other volatile compounds (Zepp et al., 1998) could lower the near surface diapycnal DOC flux, as well." We will improve the description for the diapycnal supply calculation in the revised manuscript. Page 6, lines 2-7 The "Here, we define the diapycnal supply (in mol m-3 s-1) of a solute as its vertical flux diver- gence, i.e. the change of the diapycnal flux with depth: -(âĹĞΦ_s ) ÌĚ=- $\partial/\partial z$ (Φ_S ) ÌĚ, (3)" will be edited to: "The mean diapycnal supply (-(âĹĞΦ_s ) ÌĚ, $\mu$mol kg-1 day-1) of a solute was determined as an attenuation of the diapycnal solute flux profile over depth, according to the Eq.3: -(âĹĞΦ_s ) ÌĚ=- 1/ $\partial/\partial z$ (Φ_S ) ÌĚ, (3) where  – is the in-situ density of the seawater (kg m-3), z - is depth (m) and (Φ_S ) ÌĚ (mmol m-2 day-1) – is the calculated mean diapycnal flux profile of a solute. The mean diapycnal solute supply was interpreted as the amount of a solute that is lost per unit time over a specific depth interval of the water column and was assumed to be an equivalent to microbial utilization rate of the solute. This interpretation assumes that sources other than turbulent mixing or sinks other than microbial consumption are negligible."

RC1: 6/25-26 Surface DOC concentrations >100 umol/L are not found in the ocean unless a river is nearby, which can add terrigenous DOC. The high values seem un- realistic. The values in the surface layer that are closer to 70 uM are more realistic, based on the data reported by Letscher et al. 2015 at nearby locations. The elevated DOC values at greater depth are suspect as well.

AC1: First of all, Letscher et al. (2015) does not include data collected in the nearby locations. The data, that were used in Letscher et al. (2015) for validation of the model are at least 20° off. The eastern tropical South Pacific off Peru represents a highly productive and a very dynamic coastal area with very high spatial gradients. It is influ- enced by various physical mixing processes, which are often not included in the global circulation models. Global models, in general, do not represent the upwelling regimes

well - for this, specialists use regional models. Furthermore, the near surface DOC concentrations were unlikely used for model validation. It is more likely that a mean value for depths 0-100m was used for the "euphotic zone" run. Even for those data, that have been used by Letscher et al. (2015) the model tends to underestimate or overestimate DOC concentrations, depending on the different model run. Our results, in turn, are well within the previously published range for DOC concentrations in the area off Peru. For instance, Engel and Galgani (2016) (BG) and Zäncker et al. (2017) (Front. Microbiol.) reported concentrations for DOC from 70$\mu$mol/L to 130 $\mu$mol/L for the sea surface microlayer. Franz et al. (2012) (Deep Sea Res. I) reported DOC concentrations ranging from 50 to 300 $\mu$mol/L for the upper 200m. Romankevich and Ljutsarev (1990) (Mar. Chem.) reported DOC concentrations ranging from ∼40 $\mu$mol L to ∼130 $\mu$mol L for the upper 100-150 m of the water column. Singular elevated values below surface were also reported by the same authors. Those might be related to the influence of the particle dissolution, DOM production at the deep chl a maximum (Goericke et al., 2000 (Deep Sea Res. I); Lavin et al, 2010 (Environ. Microbiol. Rep.)), etc.

---

## Author Comment (AC2) · 10 Aug 2018

We would like to thank R. Benner for his time and valuable comments to this manuscript. In the following, we will address the comments one by one.

RC2: The authors present compelling physical and biogeochemical data indicating microbial utilization of DOM plays an important role in shaping the upper oxycline in the Peruvian upwelling system. Diapycnal fluxes of O2 and DOM from productive surface waters are estimated, and analyses of DOM concentrations and compositions indicate the microbial utilization of bioavailable components (e.g. amino acids and carbohydrates) occurs mostly in the upper 50 m of the water column. In addition to the mol% compositional data presented for carbohydrates and amino acids, the DOC-normalized

yields of neutral sugars and amino acids can provide insights about the bioavailability of DOC. These data should be presented in a table (e.g. Table 1) or figure (e.g. Fig. 4).

AC2: We thank R. Benner for highlighting the interdisciplinarity of our study, which combines complex physical and biogeochemical datasets. Following his suggestion, we will add the information of DCCHO and DHAA yields (in %DOC) to Table 1 (see revised version). For better comparison with open ocean data from Kaiser and Benner (2009), we will divide our DCCHO data into neutral sugars (nS), aminosugars(S-N) and acidic sugars (S-H) and report them in $\mu$mol L-1 and in mol%DOC. The single sugar contribution to nS, S-N and S-H, will be given, as mol%nS, mol%S-N and mol%S-H, respectively. For DHAA both, mol%DOC and mol%DON will be added. GABA (mol%DHAA) will be removed from the table and will be described in the text of the reviewed manuscript as "The concentrations of GABA, which is commonly used as a signature of microbial activity (Davis et al., 2009), was very low in all samples and represented generally <1% of DHAA" (page7/line 32).

RC2: It appears carbohydrate and amino acid yields (%DOC) decline rapidly in the upper 50 m of the water column, indicating the preferential utilization of these bioavailable DOM components. The yields and bioavailability of DOC at 100 m can be compared to those at HOT and BATS to provide a more definitive #indicator of the relative bioavailability and diagenetic state of DOM at these sites.

AC: The text on page10, line 30: "A strong reworking of the labile and semi-labile DOM could also be seen from the analyses of DHAA and DCCHO composition. For instance, Glc was previously suggested to be less susceptible to microbial degradation compared to preferentially removed Fuc, Gal, and Ara (Ittekot et al., 1981; Sempere et al., 2008; Goldberg et al., 2010; Engel et al., 2012). Enrichment in Gly with depth was also previously proposed to be reflection of low nutritional value of Gly for organisms in anoxic sediments in ETSP off Chile (Pantoja and Lee, 2003) and in sediments of the North Sea (Dauwe and Middelburg, 1998). In our study, DHAA and DCCHO below

50 m depth were mainly composed by Gly and Glc, respectively, indicating a significant stage of DOM reworking. Despite the shallow depth, DOM below 50 m depth was characterized by much stronger alteration than samples collected by Kaiser and Benner (2009) from much greater depths (up to 4000m), suggesting a rapid and extensive heterotrophic DOM utilization in ETSP." will be changed to: "A strong alteration of labile and semi-labile DOM could also be seen from the analyses of DOM composition. The relatively high carbon yield (%DOC) of DHAA and DCCHO (Table 1) suggests that DOM in surface waters off Peru is more bioavailable, compared to the open ocean (Davis and Benner, 2007; Kaiser and Benner, 2009). It is, however, rapidly utilized at shallow depth. According to the classification by Davis and Benner (2007) availability of labile and semi-labile DOM in our study region was restricted to the upper 50 m of the water column. Furthermore, the compositional analyses of DHAA and DCCHO revealed that, DOM below 50 m depth was mainly composed by Gly and Glc, respectively. Glc was previously suggested to be less susceptible to microbial degradation compared to preferentially removed Fuc, Gal, and Ara (Ittekot et al., 1981; Sempere et al., 2008; Goldberg et al., 2010; Engel et al., 2012). Enrichment in Gly with depth has also been proposed to reflect the low nutritional value of Gly in anoxic sediments off Chile (Pantoja and Lee, 2003) and in sediments of the North Sea (Dauwe and Middelburg, 1998). With this, DOM in the shallow OMZ off Peru was characterized by stronger alteration compared to open ocean samples (Kaiser and Benner, 2009) at even much greater depths (up to 4000m). This suggests rapid and very extensive heterotrophic DOM utilization in the ETSP."

RC2: Observations of the low bioavailability and highly altered chemical composition of DOM at relatively shallow depths (<120 m) is likely due to upwelling of aged and altered DOM as well as active microbial utilization in surface waters (e.g. Steinfeldt et al. 2015). It appears upwelling compresses the vertical profiles of DOM concentration and composition. The manuscript would benefit from a discussion of the role of upwelling in shaping the observed biogeochemical distributions.

[Figure]

AC2: We thank R. Benner for this suggestion. The upwelling flux is likely one of the important processes governing the distribution of solutes (including DOM) in the ETSP off Peru, particularly near the coast (bottom depth less than 500m). In the revised manuscript, we will extend the discussion on upwelling (see below):

"DOM transport through the water column is achieved by advective and diffusive transport processes. Therefore, along with the turbulent mixing, other transport terms will also take their part in shaping DOM distribution off Peru. For instance, vertical advection (i.e. upwelling) of deeper waters, which are characterized by highly altered DOM in low concentrations likely contributes to a reduction of DOM concentrations near the surface layers. The upwelling driven vertical DOC flux thus counteracts the vertical turbulent diffusion and like remineralization contributes to a "compression" or sharpening of the vertical DOM concentration and composition profiles. This is unique to upwelling systems and different to the open ocean that exhibits rather smooth DOM concentration gradients and weaker changes in the DOM composition (Kaiser and Benner, 2009). Herewith, the restriction of bioavailable DOM to shallow depths by upwelling may affect the propagation depth of diapycnal DOM flux and supply. Further research is needed to improve the understanding of this interplay. "

RC2: Specific comment: The reported concentrations of the amino sugar, GalN, are very low in comparison to values in the north Pacific (HOT). The resulting GlcN:GalN ratios are extremely high (40-70). It appears there is a problem with the GalN measurements. AC2: We are thankful to R. Benner for spotting this mistake. GalN during this study was almost always below detection limit of 10nM. Table1, however included an average of these data. We will remove GalN from the table and explicitly state that the values were below detection: "S-N were represented solely by GlcN, as GalN was below DL in most samples." GlcN, could be detected in most samples, this is in accordance with GlcN:GalN ratios, typically $\geq$1.

[revised manuscript text omitted]

---

## Author Response (AR1)

Author Comment 1 (AC1) for Anonymous Review Comment 1 (RC1) on "Diapycnal dissolved organic matter supply into the Peruvian oxycline".

AC1: First of all, we are very thankful for Anonymous Referee #1 (AR1) for taking time for reading our manuscript and giving his/her comments for our study.

In the following, comments of AR1 will be addressed one by one.

RC1: **"The basic question posed in the Abstract was "how important is DOM utilization for O2 respiration within the Peruvian OMZ". The answer was not given unambiguously in the Abstract. The answer the authors should give in the Abstract, based on their results, is that "DOM introduced by vertical mixing has no role in contributing to O2 consumption in the core of the OMZ". This answer is given in the Discussion, but it is not in the Abstract. Instead, the authors state that "DOM utilization may play a significant role for shape of the upper Peruvian oxycline"; but that statement is not the answer to the question posed. The Abstract needs to be written for absolute clarity in terms of question and answer."**

AC1:

We agree with AR1 that the abstract needs to be revised for clarification of our results:

Abstract lines: 9-10

The sentence:

"However, the importance of DOM utilization for $O_2$ respiration within the Peruvian OMZ remains unclear so far."

will be changed to:

"However, the importance of DOM utilization for $O_2$ respiration in the Peruvian upwelling system in general and for shaping the upper oxycline in particular remains unclear so far."

Abstract lines: 10-16

The sentence:

"Here, we evaluate the diapycnal fluxes of $O_2$, dissolved organic carbon (DOC), dissolved organic nitrogen, dissolved hydrolysable amino acids (DHAA) and dissolved combined carbohydrates (DCCHO) and the composition of DOM in the ETSP off Peru to learn, whether labile DOM is reaching into the core of the OMZ and how important DOM utilization might be for $O_2$ attenuation."

will be changed to:

"This study reports the first estimates of diapycnal fluxes and supply of $O_2$, dissolved organic carbon (DOC), dissolved organic nitrogen, dissolved hydrolysable amino acids (DHAA) and dissolved combined carbohydrates (DCCHO) for the ETSP off Peru. Diapycnal flux and supply estimates were obtained by combining measured vertical diffusivities and solute concentration gradients. They were analysed together with the molecular composition of DCCHO and DHAA to infer the transport of labile DOM into

i

the upper OMZ and the potential role of DOM utilization for the attenuation of the diapycnal $O_2$ flux that ventilates the OMZ.”

Abstract line: 19

The line:

“suggesting that the labile DOM is already utilized”

will be changed to:

“suggesting that the labile DOM is extensively consumed”

Abstract lines: 24-25

The line:

“which suggests that DOM utilization may play a significant role for shape of the upper Peruvian oxycline.”

will be replaced with:

“which suggests that DOM utilization plays a significant role for shaping of the upper oxycline in the ETSP”

RC1: **I did not find the outcomes of this work to be enlightening. We could see in the data plots that DOC was high at the surface but low at 40, so clearly it was not surviving export by mixing to even 40 m depth. So its small (or non-existent) contribution to export into the OMZ core is pretty obvious just by looking at the distributions; the great effort by the authors to calculate vertical fluxes may have been excessive given the obvious answer to the question.**
**I'm not sure what is the main point of this paper. DOM is essentially not exported to the OMZ, but we did not need to see all the work done by the authors to know that outcome. That it contributes to the "shape of the upper oxycline" is the final finding given in the Abstract, but does that matter? The shape of the oxycline is not discussed elsewhere in the paper.**

AC1: We thank AR1 for taking the time to read our work and giving his/her opinion on our outcomes. We take this critical comment that our work is not “enlightening” as a motivation to make the importance of this first quantitative study on oxygen and DOM dynamics clearer for the reviewer and also for the reader.

We are convinced that an accurate quantification of $O_2$ and DOM fluxes and even more, the flux divergences (which are more informative for learning about sources and sinks of solutes) is an important contribution to the understanding of biogeochemical and microbial processes in OMZs and that the effort to calculate these fluxes should be valued. In particular, we do not share the statement of AR1 that “looking at the distributions” is sufficient to understand the complex $O_2$ and organic matter dynamics off Peru. The distribution of a component, determined at one time-point does not provide any information on its processing and cannot give any quantitative information on matter fluxes.

The ventilation of the OMZ by the physical supply of $O_2$ from the surface ocean is constrained by the biological utilization of oxygen for respiration of OM that happens during transport process. It has been shown that aerobic and microaerobic microbial respiration is the main pathway of organic matter remineralization (Kalvelage et al., 2015) in the oxycline. In order to estimate the attenuation of $O_2$ fluxes by microbial respiration of organic matter and by this the contribution of microbial processes to the formation and maintenance of the OMZ, we need to know the fluxes of $O_2$ and the flux attenuation (supply), i.e. $O_2$ uptake. Here, we give for the first time a quantitative estimate of those parameters and relate them (quantitatively!) to DOM supply (uptake) and to previously estimated oxygen consumption off Peru (Kalvelage et al., 2015). Please, note that the oxycline is considered as being part of the OMZ. The following sentences will be added to discussion (Page 9, line 21-25): "The observed distributions of $O_2$ and of DOC and DON components are the result of sinks and sources in the water column, mainly due to microbial processes and isopycnal and diapycnal supply (i.e. flux divergences) controlled by physical processes. A quantification of each of those individual processes is essential for understanding of important mechanisms, controlling $O_2$ and OM cycling off Peru and, therefore, the formation and maintenance of the Peruvian OMZ."

RC1: **"DOM introduced by vertical mixing has no role in contributing to O2 consumption in the core of the OMZ … DOM contributes to the "shape of the upper oxycline", but does that matter?"**

AC1: As there is no oxygen to be consumed within the core of the OMZ, organic matter, if supplied to the core of the OMZ, cannot cause oxygen consumption, unless oxygen is supplied with it. Therefore, looking at $O_2$ and DOM fluxes and their divergences is so important. Turbulent mixing is a major process for ventilating the OMZs, as ventilation by ocean currents is sluggish, and currents are carrying only low-O2 waters to the eastern boundary OMZs. Please see 9/10:" diapycnal supply is often a leading term in the flux divergence balances of $O_2$, nutrients and other solutes in the upper ocean (e.g. Schafstall et al., 2010; Kock et al., 2012; Brandt et al., 2015; Steinfeldt et al., 2015)." Our findings suggest that a substantial part of $O_2$ flux above the OMZ core is attenuated due to respiration of labile DOC. A process we refer to as "shaping of the oxycline" here.

We will clarify our description of the $O_2$ concentrations in the OMZ core in the revised manuscript:

The sentence (4/6 Methods):

"The $O_2$ optode was calibrated by a combination of Winkler titration (Winkler, 1888; Hansen, 1999) and STOX sensor measurements (Revsbech et al., 2009)."

will be changed to:

"The $O_2$ optode was calibrated by Winkler titration above the oxycline (Winkler, 1888; Hansen, 1999). The STOX sensor measurements, which revealed $O_2$ concentrations of 0.01-0.05 $\mu$mol kg$^{-1}$ within the OMZ (Revsbech et al., 2009; Thomsen et al., 2016a) were used for $O_2$ optode calibration at low $O_2$ levels."

The following will be added to the discussion (Page 12, line 25-28): "Herewith, the diapycnal supply of DHAA and DCCHO could explain up to 38% of $\overline{\nabla\Phi}_{O2}$. This suggest, that despite the diapycnal fluxes of labile and semi-labile fractions of DOM may not reach deep into the core of the OMZ, DOM based microbial respiration above the OMZ may substantially attenuate the diapycnal $O_2$ flux that ventilates the upper oxycline. In other words, DOM may alter the shape of the upper oxycline, and, therefore, contribute to the formation and maintenance of the OMZ."

RC1: **"DOM clearly not surviving export by mixing to even 40 m depth"**.

AC1: Our data represent mean values for the study area at the time of the field campaign and do not resolve episodic processes, which may occur e.g. through submesoscale mixing and supply fresh DOM into the oxycline locally (Ulloa et al., 2012, Thomsen et al., 2016b). Moreover, deepening of the mixed layer and weakening of the stratification, as for instance in Austral winter, may potentially enhance DOM supply to the deeper waters (Thomsen et al., 2016b).

The following will be added to the discussion (Page 10-11; lines 25-4): "Like for $O_2$, transport of DOM through the water column is achieved by advective and diffusive transport processes. Therefore, along with turbulent mixing, other transport terms will also take their part in shaping the DOM distribution off Peru. <…> Additionally, meso- (Thomsen et al., 2016a) and submesoscale (Thomsen et al., 2016b) dynamics have been observed in the studied area. They were shown to modify nutrient and $O_2$ distributions by stirring the water across continental slope and likely influence the DOM distribution off Peru too. However, no quantitative information on DOM fluxes, associated with upwelling, meso- or submesoscale dynamics off Peru are available to date. Seasonal and interannual variations in physical dynamics may as well affect DOM distribution off Peru, e.g. deepening of the mixed layer during Austral winter (Echevin et al., 2008) or intense downwelling/upwelling during El Niño/La Niña events (e.g. Graco et al., 2017) may result in the diapycnal DOM supply to a different depth than during typical Austral summer season."

RC1: 1/31 **"is one of the largest regions" In what regard? For an OMZ? And, "where the role of O2 concentrations discriminates." Discriminates what? And does an O2 concentration really have "a role"?**

AC1: The sentence:

"Due to the presence of a pronounced oxygen minimum zone (OMZ) (Karstensen et al., 2008), the eastern tropical South Pacific (ETSP) is one of the largest regions, where the role of $O_2$ concentrations discriminates."

will be changed to:

"The eastern tropical South Pacific (ETSP) embodies one of the largest oxygen minimum zones (OMZ) in the world ocean (Karstensen et al., 2008; Paulmier and Ruiz-Pino, 2009)."

RC1: **2/7 "anoxia-related processes" not enough information in that phrase.**

AC1: The phrase:

"anoxia-related processes (Kalvelage et al., 2013)"

will be changed to:

"anoxia-induced processes, such as denitrification (Kalvelage et al., 2011, 2013)"

RC1: **2/26: "Accessing" should be "Assessing"**

AC1: "Accessing" will be replaced with "Assessing"

RC1: **3/18 The acronym "GO" should be spelled out; presumably it is "General Oceanics"**

AC1: The line:

"Seawater was sampled with a GO rosette"

will be replaced with:

"Seawater was sampled with a rosette (GO; General Oceanics, USA)"

RC1: **6/2 What exactly is the "diapycnal solute supply"? This term should be explained fully, as it is central to the findings in the manuscript. Telling the reader that it is a 'divergence in flux' is inadequate.**

AC1: The diapycnal fluxes and supplies for all the dissolved parameters (solutes) were calculated by similar approach, therefore, the word "solutes" was used in the method for describing calculations used for dissolved oxygen, DOC, DHAA and DCCHO.

At the interface sediment-water column/air-sea interface it makes sense to speak about fluxes to quantify exchange between reservoirs. Within the water column the change of fluxes over depth (or distance), i.e. the vertical flux attenuation referred to as flux divergences, indicates the rate of consumption or production and is the value that can be compared to sources and sinks.

The flux divergence was described and calculated by equation 3. This value is an estimate for the diapycnal solute supply (an equivalent of solute consumption/remineralization), assuming that other sources or sinks (such as mesoscale, submesoscale or upwelling fluxes, or photochemical reactions) were negligible.

We will improve the description for the diapycnal supply calculation in the revised manuscript.

Page 6, lines 2-7

"Here, we define the diapycnal supply (in mol m$^{-3}$ s$^{-1}$) of a solute as its vertical flux divergence, i.e. the

change of the diapycnal flux with depth:

$$-\overline{\nabla\Phi_s} = -\frac{\partial}{\partial z}\overline{\Phi_S},$$ (3)"

will be edited to:

"The mean diapycnal supply ($-\overline{\nabla\Phi_S}$, µmol kg$^{-1}$ day$^{-1}$) of a solute was determined at 28 m depth intervals as an attenuation of the diapycnal solute flux profile over depth, according to the Eq. 3:

$$-\overline{\nabla\Phi_s} = -\frac{1}{\rho}\frac{\partial}{\partial z}\overline{\Phi_S},$$ (3)

where $\rho$ – is the *in-situ* density of the seawater (kg m$^{-3}$), z - is depth (m) and $\overline{\Phi_S}$ (mmol m$^{-2}$ day$^{-1}$) – is the estimated mean diapycnal flux profile of a solute. The mean diapycnal solute supply was interpreted to balance the amount of a solute that is lost per unit of time over a specific depth interval of the water column due to the microbial utilization of the solute. This interpretation assumes that sources other than turbulent mixing or sinks other than microbial consumption are negligible."

RC1: **6/25-26 Surface DOC concentrations >100 umol/L are not found in the ocean unless a river is nearby, which can add terrigenous DOC. The high values seem unrealistic. The values in the surface layer that are closer to 70 uM are more realistic, based on the data reported by Letscher et al. 2015 at nearby locations. The elevated DOC values at greater depth are suspect as well.**

AC1: First of all, Letscher et al. (2015) does not include data collected in the nearby locations. The data, that were used in Letscher et al. (2015) for validation of the model are at least 20° off.

The eastern tropical South Pacific off Peru represents a highly productive and a very dynamic coastal area with very high spatial gradients. It is influenced by various physical mixing processes, which are often not included in the global circulation models. Global models, in general, do not represent the upwelling regimes well - for this, specialists use regional models. Furthermore, the near surface DOC concentrations were unlikely used for model validation. It is more likely that a mean value for depths 0-100m was used for the "euphotic zone" run.

Even for those data, that have been used by Letscher et al. (2015) the model tends to underestimate or overestimate DOC concentrations, depending on the different model run.

Our results, in turn, are well within the previously published range for DOC concentrations in the area off Peru. For instance, Engel and Galgani (2016) (BG) and Zäncker et al. (2017) (Front. Microbiol.) reported concentrations for DOC from 70µmol/L to 130 µmol/L for the sea surface microlayer. Franz et al. (2012) (Deep Sea Res. I) reported DOC concentrations ranging from 50 to 300 µmol/L for the upper 200m. Romankevich and Ljutsarev (1990) (Mar. Chem.) reported DOC concentrations ranging from ~40 µmol L to ~130 µmol L for the upper 100-150 m of the water column. Singular elevated values below surface

were also reported by the same authors. Those might be related to the influence of the particle dissolution, DOM production at the deep chl a maximum (Goericke et al., 2000 (Deep Sea Res. I); Lavin et al, 2010 (Environ. Microbiol. Rep.)), etc.

However, we may not fully exclude possible contamination, therefore the following will be added to Results (Page7, lines 21-25): "DOC concentrations of >100 µmol L$^{-1}$ had been reported previously for the water column off Peru (Romankevich and Ljutsarev, 1990; Franz et al., 2012a). However, since concentrations >100 µmol L$^{-1}$ were observed only sporadically, we cannot exclude a possible contamination of these samples."

Author Comment 2 (AC2) for Review Comment 2 (RC2) by Prof. Dr. R. Benner on "Diapycnal dissolved organic matter supply into the Peruvian oxycline".

We would like to thank R. Benner for his time and valuable comments to this manuscript.

In the following, we will address the comments one by one.

RC2: **The authors present compelling physical and biogeochemical data indicating microbial utilization of DOM plays an important role in shaping the upper oxycline in the Peruvian upwelling system. Diapycnal fluxes of O2 and DOM from productive surface waters are estimated, and analyses of DOM concentrations and compositions indicate the microbial utilization of bioavailable components (e.g. amino acids and carbohydrates) occurs mostly in the upper 50 m of the water column.**

**In addition to the mol% compositional data presented for carbohydrates and amino acids, the DOC-normalized yields of neutral sugars and amino acids can provide insights about the bioavailability of DOC. These data should be presented in a table (e.g. Table 1) or figure (e.g. Fig. 4).**

AC2: We thank R. Benner for highlighting the interdisciplinarity of our study, which combines complex physical and biogeochemical datasets. Following his suggestion, we will add the information of DCCHO and DHAA yields (in %DOC) to Table 1:

Table 1: Relative composition (mol%) of dissolved hydrolysable amino acids (DHAA) and dissolved combined carbohydrates (DCCHO) in the water column, "n.d." - not detectable. The DCCHO are divided into three classes, nS – neutral sugars, S-N – amino-sugars, and S-H – acidic sugars. The number of samples at each depth interval, used for calculation of the average value, is given as "n". The mean values for DHAA and DCCHO composition below the mixed layer (10 to 122 m) are reported for similar depth intervals (14 m) as diapycnal DOM and O2 fluxes. The mean values for DHAA and DCHO within the mixed layer are reported for ~5 m depth intervals.

| Depth (m) | n | DHAA $(\mu mol\ L^{-1})$ | DHAA (%DOC) | DHAA (%DON) | mol% DHAA Gly | Thr | Ala | Asp | Glu | Ser | Arg | Leu | Val | Ileu | Phe | Tyr |
|---|---|---|---|---|---|---|---|---|---|---|---|---|---|---|---|---|
| 1-5 | 30 | 0.6±0.3 | 2±1 | 15±10 | 22±4 | 9±1 | 11±1 | 17±1 | 15±3 | 11±2 | 2.3±0.3 | 4±1 | 3.0±0.4 | 2.5±0.6 | 2.4±0.4 | 1.8±0.4 |
| 5-10 | 25 | 0.5±0.3 | 2.3±0.9 | 15±9 | 23±4 | 9±2 | 11±1 | 17±1 | 15±4 | 10±1 | 2.2±0.4 | 4±1 | 2.9±0.6 | 2.1±0.5 | 2.1±0.4 | 1.7±0.3 |
| 10-24 | 48 | 0.4±0.2 | 1.8±0.8 | 16±14 | 25±4 | 9±2 | 11±1 | 17±1 | 13±2 | 9±1 | 2.1±0.6 | 3±1 | 2.8±0.7 | 2.2±0.7 | 2.1±0.6 | 1.9±0.5 |
| 24-38 | 28 | 0.24±0.07 | 1.2±0.3 | 12±14 | 28±3 | 10±1 | 12±1 | 17±1 | 11±2 | 9±1 | 1.9±0.4 | 3±1 | 2.4±0.6 | 1.9±0.6 | 1.8±0.4 | 2.0±0.7 |
| 38-52 | 34 | 0.20±0.05 | 1.0±0.4 | 9±7 | 29±6 | 10±2 | 12±2 | 16±2 | 11±2 | 9±1 | 1.8±0.6 | 3±2 | 2.3±0.8 | 1.7±0.5 | 1.8±0.4 | 1.7±0.4 |
| 52-66 | 35 | 0.17±0.03 | 0.9±0.3 | 13±19 | 31±3 | 10±2 | 12±1 | 16±1 | 10±2 | 8±1 | 1.7±0.4 | 2±1 | 2.4±0.5 | 1.6±0.7 | 1.7±0.3 | 1.7±0.5 |
| 66-80 | 27 | 0.16±0.05 | 0.9±0.3 | 9±8 | 32±4 | 10±1 | 12±1 | 15±2 | 10±2 | 8±1 | 1.7±0.5 | 2±1 | 2.5±0.6 | 1.7±0.8 | 1.7±0.4 | 1.8±0.5 |
| 80-94 | 22 | 0.15±0.08 | 0.9±0.4 | 8±7 | 34±3 | 10±2 | 12±2 | 15±1 | 10±2 | 9±1 | 1.6±0.4 | 2±1 | 2.2±0.7 | 1.3±0.7 | 1.6±0.4 | 1.6±0.4 |
| 94-108 | 14 | 0.13±0.03 | 0.7±0.2 | 9±8 | 34±3 | 10±2 | 13±2 | 15±2 | 9±2 | 8±2 | 1.6±0.5 | 2±1 | 2.3±0.7 | 2±1 | 1.7±0.4 | 1.7±0.9 |
| 108-122 | 13 | 0.13±0.03 | 0.8±0.2 | 6±4 | 32±3 | 10±2 | 12±1 | 16±2 | 10±2 | 8±1 | 1.7±0.3 | 3±1 | 2.3±0.8 | 2±1 | 1.9±0.4 | 1.7±0.5 |
| 122-200 | 18 | 0.12±0.03 | 0.7±0.3 | 8±6 | 35±3 | 10±1 | 12±2 | 15±2 | 9±1 | 8±2 | 1.5±0.7 | 2±1 | 2.5±0.6 | 1.7±0.7 | 1.5±0.4 | 1.5±0.5 |

| Depth (m) | n | DCCHO $(\mu mol\ L^{-1})$ nS | S-N | S-H | mol%DOC nS | S-N | S-H | mol% nS Glc | ManXyl | Gal | Rhm | Fuc | Ara | mol% S-H Glu-URA | Gal-URA | Glc-H |
|---|---|---|---|---|---|---|---|---|---|---|---|---|---|---|---|---|
| 1-5 | 30 | 1.5±0.8 | 0.10±0.03 | 0.10±0.08 | 9±4 | 0.6±0.2 | 0.6±0.3 | 30±13 | 32±6 | 17±6 | 11±8 | 8±2 | 2±1 | 48±21 | 51±21 | 0.4±2 |
| 5-10 | 25 | 1.1±0.6 | 0.08±0.03 | 0.07±0.05 | 8±4 | 0.5±0.1 | 0.5±0.3 | 33±11 | 33±5 | 16±6 | 8±6 | 8±2 | 2±1 | 43±26 | 55±24 | 2±10 |
| 10-24 | 47 | 0.7±0.3 | 0.06±0.02 | 0.04±0.03 | 5±2 | 0.4±0.1 | 0.3±0.2 | 36±13 | 37±8 | 12±5 | 5±4 | 7±2 | 2±1 | 32±25 | 67±25 | 1±7 |
| 24-38 | 28 | 0.4±0.1 | 0.04±0.01 | 0.02±0.02 | 4±1 | 0.3±0.1 | 0.2±0.1 | 43±11 | 38±7 | 9±4 | 2±2 | 6±2 | 0.4±1.0 | 20±20 | 80±20 | n.d. |
| 38-52 | 35 | 0.4±0.2 | 0.03±0.01 | 0.02±0.01 | 4±2 | 0.3±0.1 | 0.1±0.1 | 42±10 | 41±9 | 9±3 | 2±2 | 5±2 | 0.3±0.8 | 28±30 | 72±30 | n.d. |
| 52-66 | 34 | 0.5±0.2 | 0.03±0.01 | 0.02±0.02 | 4±2 | 0.2±0.1 | 0.2±0.2 | 45±9 | 41±9 | 7±4 | 2±2 | 5±2 | 0.2±0.6 | 21±27 | 77±28 | 2±11 |
| 66-80 | 27 | 0.4±0.2 | 0.02±0.01 | 0.01±0.01 | 4±2 | 0.2±0.1 | 0.1±0.1 | 47±13 | 44±12 | 5±3 | 1±1 | 3±2 | 0.3±0.7 | 19±28 | 81±28 | n.d. |
| 80-94 | 22 | 0.4±0.2 | 0.02±0.01 | 0.01±0.01 | 4±2 | 0.2±0.1 | 0.1±0.1 | 47±11 | 45±10 | 4±3 | 0.1±0.6 | 2±2 | 0.7±1.3 | 32±33 | 68±33 | n.d. |
| 94-108 | 15 | 0.3±0.1 | 0.02±0.01 | 0.01±0.01 | 3±1 | 0.2±0.1 | 0.1±0.1 | 53±11 | 40±10 | 4±3 | 0.1±0.5 | 2±2 | 0.2±0.9 | 28±29 | 72±29 | n.d. |
| 108-122 | 13 | 0.4±0.1 | 0.02±0.01 | 0.02±0.02 | 4±2 | 0.2±0.1 | 0.2±0.2 | 51±16 | 43±14 | 3±3 | 0.2±0.7 | 2±2 | 0.3±1.0 | 44±46 | 56±46 | n.d. |
| 122-200 | 18 | 0.4±0.2 | 0.02±0.01 | 0.01±0.02 | 4±1 | 0.2±0.1 | 0.1±0.2 | 52±10 | 44±9 | 2±2 | n.d. | 1±2 | 0.7±2.3 | 22±30 | 78±30 | n.d. |

For better comparison with open ocean data from Kaiser and Benner (2009), we will divide our DCCHO data onto neutral sugars (nS), aminosugars(S-N) and acidic sugars (S-H) and report them in µmol L$^{-1}$ and in mol%DOC. The single sugar contribution to nS, S-N and S-H, will be given, as mol%nS, mol%S-N and mol%S-H, respectively. For DHAA both, mol%DOC and mol%DON will be added.

GABA (mol%DHAA) will be removed from the table and will be described in the text of the reviewed manuscript as "The concentrations of GABA, which is commonly used as a signature of microbial activity (Davis et al., 2009), was very low in all samples and represented generally <1% of DHAA" (page8/line 11).

RC2: **It appears carbohydrate and amino acid yields (%DOC) decline rapidly in the upper 50 m of the water column, indicating the preferential utilization of these bioavailable DOM components. The yields and bioavailability of DOC at 100 m can be compared to those at HOT and BATS to provide a more definitive #indicator of the relative bioavailability and diagenetic state of DOM at these sites.**

AC:  The following will be added to discussion (page12, lines 1-13):

"As DHAA and DCCHO are preferentially utilized during microbial decomposition of OM (Skoog and Benner, 1997; Lee et al., 2000; Amon et al., 2001), their carbon yield (%DOC) and composition may serve as indicators of diagenetic history of DOM (e.g. Kaiser and Benner, 2009; Davis et al., 2009). Thus, the relatively high carbon yield of DHAA and DCCHO (Table 1), found near the surface during our study, suggests that DOM in surface waters off Peru is more bioavailable, compared to the open ocean (Davis and Benner, 2007; Kaiser and Benner, 2009). It is, however, rapidly altered at shallow depth. Applying the classification of Davis and Benner (2007), that implies that carbon yields of DHAA above 1.6 %DOC and 1.09 %DOC are corresponding to labile and semi-labile DOM, respectively, to our data suggests that the labile and semi-labile DOM off Peru was restricted upper 50 m of the water column."

RC2: **Observations of the low bioavailability and highly altered chemical composition of DOM at relatively shallow depths (<120 m) is likely due to upwelling of aged and altered DOM as well as active microbial utilization in surface waters (e.g. Steinfeldt et al. 2015). It appears upwelling compresses the vertical profiles of DOM concentration and composition. The manuscript would benefit from a discussion of the role of upwelling in shaping the observed biogeochemical distributions.**

AC2: We thank R. Benner for this suggestion. The upwelling flux is likely one of the important processes governing the distribution of solutes (including DOM) in the ETSP off Peru, particularly near the coast (bottom depth less than 500m).

In the revised manuscript, we will extend the discussion on upwelling (Page 10, lines 25-33):

"Like for $O_2$, transport of DOM through the water column is achieved by advective and diffusive transport processes. Therefore, along with turbulent mixing, other transport terms will also take their part in shaping the DOM distribution off Peru. For instance, vertical advection (i.e. upwelling) transports deep water, which is characterized by highly altered DOM and low DOC concentrations, into the upper ocean near the continental margins. The upwelling may counteract the turbulent downward flux of DOC and, therefore, contribute to a "compression" or sharpening of the vertical DOM concentration and composition profiles. This is unique to upwelling systems and different to the open ocean regions where low DOC concentration gradients and smaller changes in the DOM composition were observed at similar depth (Kaiser and Benner, 2009). "

(Page 12, lines 15-23):

"Therewith, our data suggest that DOM in the shallow OMZ off Peru was characterized by stronger alteration compared to open ocean samples (Kaiser and Benner, 2009) at even much greater depths (up to 4000m). This may be due to both, an upwelling of altered DOM from the deep and a rapid and very extensive heterotrophic DOM utilization in the ETSP. The upwelling may "compress" labile and semi-labile DOM towards the surface, while the rapid microbial utilization of DOM shall prevent labile and semi-labile DOM export into the OMZ, and also would imply a pronounced heterotrophic respiration"

RC2: **Specific comment:**

**The reported concentrations of the amino sugar, GalN, are very low in comparison to values in the north Pacific (HOT). The resulting GlcN:GalN ratios are extremely high (40-70). It appears there is a problem with the GalN measurements.**

AC2: We are thankful to Dr. R. Benner for spotting this mistake. GalN during this study was almost always below detection limit of 10nM. Table1, however included an average of these data. We will remove GalN from the table and explicitly state that the values were below detection (Page 8, line 2):

"S-N were represented solely by GlcN, as GalN was below DL in most samples."

GlcN, could be detected in most samples, this is in accordance with GlcN:GalN ratios, typically $\geq$1.

[revised manuscript text omitted]
$^{-1}$) nS | S-N | S-H | mol%DOC nS | S-N | S-H | mol% nS Glc | ManXyl | Gal | Rhm | Fuc | Ara | mol% S-H Glu-URA | Gal-URA | Glc-H |
|---|---|---|---|---|---|---|---|---|---|---|---|---|---|---|---|---|
| 1-5 | 30 | 1.5±0.8 | 0.10±0.03 | 0.10±0.08 | 9±4 | 0.6±0.2 | 0.6±0.3 | 30±13 | 32±6 | 17±6 | 11±8 | 8±2 | 2±1 | 48±21 | 51±21 | 0.4±2 |
| 5-10 | 25 | 1.1±0.6 | 0.08±0.03 | 0.07±0.05 | 8±4 | 0.5±0.1 | 0.5±0.3 | 33±11 | 33±5 | 16±6 | 8±6 | 8±2 | 2±1 | 43±26 | 55±24 | 2±10 |
| 10-24 | 47 | 0.7±0.3 | 0.06±0.02 | 0.04±0.03 | 5±2 | 0.4±0.1 | 0.3±0.2 | 36±13 | 37±8 | 12±5 | 5±4 | 7±2 | 2±1 | 32±25 | 67±25 | 1±7 |
| 24-38 | 28 | 0.4±0.1 | 0.04±0.01 | 0.02±0.02 | 4±1 | 0.3±0.1 | 0.2±0.1 | 43±11 | 38±7 | 9±4 | 2±2 | 6±2 | 0.4±1.0 | 20±20 | 80±20 | n.d. |
| 38-52 | 35 | 0.4±0.2 | 0.03±0.01 | 0.02±0.01 | 4±2 | 0.3±0.1 | 0.1±0.1 | 42±10 | 41±9 | 9±3 | 2±2 | 5±2 | 0.3±0.8 | 28±30 | 72±30 | n.d. |
| 52-66 | 34 | 0.5±0.2 | 0.03±0.01 | 0.02±0.02 | 4±2 | 0.2±0.1 | 0.2±0.2 | 45±9 | 41±9 | 7±4 | 2±2 | 5±2 | 0.2±0.6 | 21±27 | 77±28 | 2±11 |
| 66-80 | 27 | 0.4±0.2 | 0.02±0.01 | 0.01±0.01 | 4±2 | 0.2±0.1 | 0.1±0.1 | 47±13 | 44±12 | 5±3 | 1±1 | 3±2 | 0.3±0.7 | 19±28 | 81±28 | n.d. |
| 80-94 | 22 | 0.4±0.2 | 0.02±0.01 | 0.01±0.01 | 4±2 | 0.2±0.1 | 0.1±0.1 | 47±11 | 45±10 | 4±3 | 0.1±0.6 | 2±2 | 0.7±1.3 | 32±33 | 68±33 | n.d. |
| 94-108 | 15 | 0.3±0.1 | 0.02±0.01 | 0.01±0.01 | 3±1 | 0.2±0.1 | 0.1±0.1 | 53±11 | 40±10 | 4±3 | 0.1±0.5 | 2±2 | 0.2±0.9 | 28±29 | 72±29 | n.d. |
| 108-122 | 13 | 0.4±0.1 | 0.02±0.01 | 0.02±0.02 | 4±2 | 0.2±0.1 | 0.2±0.2 | 51±16 | 43±14 | 3±3 | 0.2±0.7 | 2±2 | 0.3±1.0 | 44±46 | 56±46 | n.d. |
| 122-200 | 18 | 0.4±0.2 | 0.02±0.01 | 0.01±0.02 | 4±1 | 0.2±0.1 | 0.1±0.2 | 52±10 | 44±9 | 2±2 | n.d. | 1±2 | 0.7±2.3 | 22±30 | 78±30 | n.d. |

**Table 2: Diapycnal fluxes and supplies (in bold) of $O_2$ and DOM: DOC, DON, dissolved organic carbon in DCCHO and DHAA and dissolved organic nitrogen in DHAA 95% confidence intervals, calculated after Schafstall et al. (2010) for each parameter, are presented in brackets. BLM – "below the mixed layer" – a depth, defined below 10m of the water column, using a threshold criterion of 0.2°C temperature decrease.**

[revised manuscript text omitted]

Variables factor map (PCA)

Individuals factor map (PCA)

---

## Author Response (AR3)

The answer for the review comments for BG-2018-284.

In the following the authors will address review comments one by one. the review comment will be marked as R1 or R2, for reviewer1 and 2, respectively. Authors' response will be marked as A.

General Comments:

R1:     1. The Introduction needs to be better written. I was confused by the logical construction of the main points being made too many times. For example, the statement "changes in the remineralization rate of DOM might rather be linked to lack of bioavailable OM supply into the OMZ than to low-O2 conditions" was not preceded by any mention that DOM consumption rates vary in the water column. Instead, we were told that "Recent studies in the upwelling area and the corresponding OMZ off Chile… found bacterial activity of similar range to the oxygenated waters." This finding does not tell us about DOM.

A:      We thank the R1 for his/her comments.

The sentence (page 2 line 20): "Thus, the rapid microbial decomposition of labile organic matter in the euphotic zone is commonly followed by slower decomposition of less bioavailable semi-labile DOM and very slow decomposition of extensively reworked refractory DOM deeper the water column (e.g. Hansell, 2013)." Was added to the revised version of the manuscript.

The paragraph (page 2, line 25-34) was rephrased into: "Microbial decomposition of organic matter has previously been suggested to be limited under anoxia (Harvey et al., 1995; Nguyen and Harvey, 1997). Following this suggestion, one may assume that, if labile DOM is supported to the OMZ, it would not be reworked as rapidly as in oxygenated waters. Recent studies in the upwelling area and the corresponding OMZ off Chile found, however, that even under anoxia the ability of microbes to decompose labile DOM (Leucine-incorporation rate) did not differ from the oxygenated waters (Sempéré et al., 2008, Pantoja et al., 2009). These studies suggest that slower remineralization of DOM in OMZ might rather be caused by lack of bioavailable organic matter supply into the OMZ than by low-$O_2$ conditions. Herewith, measured concentrations of bioavailable components of DOM over the water column in the ETSP are yet controversary. For instance, Pantoja et al. (2009) reported relatively high concentrations of free and combined AA in the OMZ off Chile. Sempere et al. (2008) reported low concentrations of neutral CHO in the corresponding upwelling area, compared to the open Pacific Ocean."

R1: Another example of confusing text: "Assessing the possible effects of low-O2 conditions on the composition of DOM implies that the DOM is transported into the OMZ from the oxygenated waters." I do not know what that means.

A: The sentence has been removed.

R1: I am not a fan of averaging all of the measurements (as in Figure 4) and then evaluating the profiles of the mean concentrations for vertical fluxes. Clearly, the waters very near the coast are upwelling while those further offshore are highly stratified. In O2 (Fig 4a), for example, we see surface concentrations ranging from near zero at the coast to >200 uM offshore; the mean of O2 is neither useful nor insightful. I suggest evaluating the stratified region for diapyncal fluxes. Inclusion of the waters that are actually upwelling will bias the answer to the question posed. The authors agree with the when they state that "This spatial averaging is likely responsible for a lower near-surface diapycnal O2 flux compared to other eastern boundary upwelling systems", so why do the averaging?

A: We thank the R1 for the comment. We would like to draw the reviewer's attention to the chapter 2.3, and that the diapycnal fluxes were calculated using a solutes profile and the microstructure profiles collected directly afterwards or before this individual profile. Averaging was then done for the fluxes to get one mean flux profile and further calculating of the diapycnal solute supply. The Figure 4, where the mean distribution is shown, is used only for easier visualization of the solute's gradients and only for the sake of discussion.

Following the reviewer's suggestion, we have added the separated presentation of average solute's profiles on the onshore (<40 km) and offshore (>40 km) to the Figure 4. We have added a discussion of the onshore and offshore diapycnal oxygen fluxes and supply (page 8, line 14):

"Onshore (<40 km) and offshore (>40 km) $O_2$ fluxes did not differ statistically. This likely was due to the fact that while vertical oxygen gradients were enhanced in the offshore region (Fig.4), the turbulence and, thus, eddy diffusivities were elevated in the onshore region."

and on page 8 line 17:

"Again, onshore (<40 km) and offshore (>40km) the diapycnal $O_2$ supply was not statistically different."

However, we refrained from presenting numbers of diapycnal fluxes and supply of $O_2$, DOC, DON, DCCHO and DHAA for the onshore and offshore regions separately. This is because the 95% confidence intervals for the regionally separated fluxes become very unfavourable. The elevated uncertainty originates from the elevated temporal variability of turbulence in the ocean. Diapycnal fluxes of solutes in the ocean are driven by turbulent

mixing events that are short-lived, patchy and occur sporadically. With time, elevated mixing events are distributed near-logarithmically and very strong events, having 10000 (10^4) times more energy than weak events and enhancing turbulent mixing by the same order of magnitude, occur very infrequent. Yet, it is the strength and the frequency of occurrence of these elevated mixing events that determines the magnitude of the diapycnal flux of DOM or of oxygen. The variability of the turbulence is reflected in the elevated spread of the upper and lower 95% confidence limits presented in the Table 2. Thus, in order to estimate a representative diapycnal flux of $O_2$ and DOM we had to combine all data available from the study area to determine a single diapycnal solute flux and supply profile. This study uses 263 turbulence profiles collected in the study area along with respective oxygen and DOM profiles. Each turbulence profile collected during the cruise required about 20 min of the ship time. Thus, the used data set represents more than 3.5 days of continuous turbulence measurements. Any separation of the dataset would strongly degrade statistical confidence of the calculated fluxes and supply.

Certainly, we expect regional differences within the study area. However, a different measurement program would need to be designed focusing solely on those differences. In our contribution, we intend to provide the first estimates of average diapycnal fluxes of $O_2$, DOC, DON, DCCHO and DHAA in the Peruvian upwelling region aiming at advancing the understanding of the prominent biogeochemical processes.

R:     I'm unclear on why the authors offered in this paper two disconnected questions: i) how important is the diapycnal flux of DOM and ii) does the low O2 in the OMZ cause changes in the DOM composition?

A:     As the cycling of DOM in the ETSP was little studied prior to our cruise, our research questions were based on the limited information that we had and the general understanding of the DOM cycling and the OMZ.

The ETSP is characterized by high primary production (which could potentially lead to high concentrations of DOM) in surface waters over a permanent OMZ in the subsurface layer. Those characteristics motivate our first research question about whether bioavailable DOM may be supplied to the upper OMZ via turbulent mixing.

Consequently, the first question then led to the investigation of related scientific issues, such as: How strong the $O_2$ supply (which is supposed to ventilate upper oxycline of the OMZ) is actually reduced due to utilization of the DOM/or how important is DOM for sustaining of the OMZ? How would DOM cycling be altered in the upper OMZ? Will the reworking of DOM slow down? How it will affect the composition of DOM?

We formulated those question prior to our work and to access those questions we combined two approaches: physical flux measurements and analyses of DOM composition. We estimated the diapycnal supplies of solutes, compared the supply of DOM to supply of $O_2$ and also compared the composition of DOM from OMZ influenced water column to the composition of DOM from well oxygenated waters. In the Introduction section we discuss possible effects of anoxia on DOM cycling to provide context to our scientific questions.

Herewith, our very different approaches agreed well, and the results of our study suggested that DOM is extensively reworked already above the upper oxycline by aerobic respiration; thus, DOM may be supplied to the OMZ is in already highly reworked stage. However, the remineralization of DOM above the upper oxycline may substantially attenuate $O_2$ supply that ventilates OMZ.

R1: Specific Comments:

Page/Line

R1:     3/20: the "L" is missing in the units for P.

A:      "μmo" was replaced with "μmol"

R1:     3/22: "where" rather than "here"

A:      "here" was replaced with "where"

R1:     5/1: remove "," after "significant". In general, the authors should look at every comma used and determined whether or not it is used correctly; there were too many poorly placed commas in the text. Do not use them where a pause is not intended.

A:      The comma was removed. We also removed not needed commas throughout the text.

R1:     7/11: I appreciate the author's doubting the very high DOC values they report. DOC concentrations of 100-120 uM are not typically observed unless a river is nearby. Franz et al 2012 was cited as a paper similarly reporting high DOC off Peru, but that work was done in a mesocosm; I do not see where the authors reported high DOC in a natural setting. And the DOC data coming out of the Romankevich lab in the 80's, another paper cited for high values off Peru, are also highly suspect. I wouldn't cite that paper since the data are not reliable. I hope these high values were not included in calculating the diapycnal fluxes of DOC; the high values should be excluded.

A:      We thank the R1 for this comment. However, the R1 refers to Franz et al 2012b – paper in the "Biogeosciences" Journal. For the DOC, we cited Franz et. al. 2012a in the Journal "Deep Sea Research Part I". This paper contains field DOC data. The DOC concentrations reported by Franz et al 2012a vary from ~50 to 300 µmol/L. Therefore, our data are within the reported range.

R1:      7/20: "DCCHO represented from 1 to 25% of DOC". I am confused about which DCCHO data to look at when considering that statement. In Table 1 I see the DCCHO in the depth range of 1-5 m sums to 1.7 umol/L (nS+S-N+S-H), while in Fig. 4d the DCCHO mean is closer to 10 umol/L. Which DCCHO data should I be looking at; the two sources (table vs figure) seem to disagree with one another.

A:      We thank the R1 for the comment. Please, note that the data presented in the figure 4 is expressed in µmol C/L for DCCHO and DHAA and in µmol N/ L for DHAA, while the values in the Table 1 are expressed in µmol monomer /L of DCCHO and DHAA.

R1:      Table 2: why are the numbers in bold type? I do not see the need for that. And why in the lower section is the upper most depth range "BML-38", which reaches into the middle of the depth range just below it (24-52). Should it be "BML-24", as in the upper section of the table?

A:      We thank the R1 for the comment. The bold type was changed to the plain text.

We would like to draw the attention of the R1 that the diapycnal supply was obtained by calculating the difference between two flux values (and dividing by the depth distance and the density) as described in the chapter 2.3. Therefore, each supply value is based on the two flux estimates, resulting in the larger, shifted and overlapping depth ranges.

R1:      8/21: remove ',' after "to understand"

A:      The comma has been removed

Report #2

R2:     This study provides novel and interesting observations of dissolved oxygen and DOM dynamics in the upper water column of the highly productive Peru upwelling system. The microbial utilization of DOM and dissolved oxygen plays an important role in shaping the upper oxycline and maintaining the oxygen minimum zone. The carbohydrate and amino acid components of DOM were preferentially utilized in the upper water column, indicating rapid utilization of labile and semi-labile components of DOM. The composition of DOM in the oxygen minimum zone indicated extensive alteration and the removal of most bioavailable components.

R2:     I think the authors should consider particles as potentially important sources of DOM in the upper water column. Particulate matter is abundant in this system and the microbial utilization of POM is a major sink for dissolved oxygen in the upper water column. Microbial degradation of POM also releases DOM and is a potentially important source of labile and semi-labile components of DOM (Smith et al. 1992). Therefore, the diapycnal flux approach in this study likely underestimates the role of microbial respiration of DOM in shaping the upper oxycline.

A:      We thank R. Benner for the comment.
        We added following paragraph to the Discussion (page 10 Lines 23-29):
        "DOM might also be transported to depth within particles. Thus, the "uncoupled" dissolution of large sinking aggregates as a result of bacterial enzymatic activity (Smith et al., 1992) or abiotically (Sempéré et al., 2000) may serve as an additional DOM source and, therefore, affect the distribution of DOM in the water column. The sporadic dissolution of particles may bias the diapycnal DOM flux estimates at individual stations. Therefore, the bias may be reduced by calculating the mean diapycnal flux over a large number of depth profiles. The continuous DOM release from POM over the water column (e.g. Lefèvre et al., 1996), in turn, may lead to an overestimation of diapycnal DOM fluxes and DOM based microbial respiration. However, no direct measurements of the DOM fraction resulting from particle dissolution exist so far in the studied area."

Specific comments:

R2:     The authors use many abbreviations, some of which are not common. This makes reading the article tedious in some areas, and I recommend eliminating uncommon abbreviations (e.g. n-S, S-H, S-N, DURA, etc.).

A:      The abbreviations nS, S-H, S-N and DURA have been removed from the revised version of the text, the abbreviations for the individual analysed acidic, uronic and amino sugars were replaced according to the abbreviation list given in FEBS Journal Volume 74, pp 1-6, 1977

(doi:10.1111/j.1432-1033.1977.tb11359.x.). Acronyms nS, SN and SA for neutral sugars, amino sugars and acidic sugars, respectively, however had to be kept in the Table1 for the sake of space. Those acronyms are introduced in the Table's legend.

R2:     Pg 5, line 7: The amino acids asparagine and glutamine are deaminated during acid hydrolysis, and they contribute to the measured concentrations of aspartic acid and glutamic acid, respectively.

A:     We added the following sentence to the Page 5 Line 14:

[revised manuscript text omitted]

---

## Author Response (AR4)

Dear Editor, lieber Gerhard,

We are very glad to hear about the final acceptance of our manuscript. For the final text version we have adopted the technical suggestions provided by referee 1, as also shown in the marked up manuscript. We also added the link to the database (PANGEA) to show where the data will become publicly available.

Best regards,

Anja Engel

[revised manuscript text omitted]
$^{-1}$) nS | SN | SA | mol%DOC nS | SN | SA | mol% nS Glc | ManXyl | Gal | Rhm | Fuc | Ara | mol% SA GluUA | GalUA | GlcA |
|---|---|---|---|---|---|---|---|---|---|---|---|---|---|---|---|---|
| 1-5 | 30 | 1.5±0.8 | 0.10±0.03 | 0.10±0.08 | 9±4 | 0.6±0.2 | 0.6±0.3 | 30±13 | 32±6 | 17±6 | 11±8 | 8±2 | 2±1 | 48±21 | 51±21 | 0.4±2 |
| 5-10 | 25 | 1.1±0.6 | 0.08±0.03 | 0.07±0.05 | 8±4 | 0.5±0.1 | 0.5±0.3 | 33±11 | 33±5 | 16±6 | 8±6 | 8±2 | 2±1 | 43±26 | 55±24 | 2±10 |
| 10-24 | 47 | 0.7±0.3 | 0.06±0.02 | 0.04±0.03 | 5±2 | 0.4±0.1 | 0.3±0.2 | 36±13 | 37±8 | 12±5 | 5±4 | 7±2 | 2±1 | 32±25 | 67±25 | 1±7 |
| 24-38 | 28 | 0.4±0.1 | 0.04±0.01 | 0.02±0.02 | 4±1 | 0.3±0.1 | 0.2±0.1 | 43±11 | 38±7 | 9±4 | 2±2 | 6±2 | 0.4±1.0 | 20±20 | 80±20 | n.d. |
| 38-52 | 35 | 0.4±0.2 | 0.03±0.01 | 0.02±0.01 | 4±2 | 0.3±0.1 | 0.1±0.1 | 42±10 | 41±9 | 9±3 | 2±2 | 5±2 | 0.3±0.8 | 28±30 | 72±30 | n.d. |
| 52-66 | 34 | 0.5±0.2 | 0.03±0.01 | 0.02±0.02 | 4±2 | 0.2±0.1 | 0.2±0.2 | 45±9 | 41±9 | 7±4 | 2±2 | 5±2 | 0.2±0.6 | 21±27 | 77±28 | 2±11 |
| 66-80 | 27 | 0.4±0.2 | 0.02±0.01 | 0.01±0.01 | 4±2 | 0.2±0.1 | 0.1±0.1 | 47±13 | 44±12 | 5±3 | 1±1 | 3±2 | 0.3±0.7 | 19±28 | 81±28 | n.d. |
| 80-94 | 22 | 0.4±0.2 | 0.02±0.01 | 0.01±0.01 | 4±2 | 0.2±0.1 | 0.1±0.1 | 47±11 | 45±10 | 4±3 | 0.1±0.6 | 2±2 | 0.7±1.3 | 32±33 | 68±33 | n.d. |
| 94-108 | 15 | 0.3±0.1 | 0.02±0.01 | 0.01±0.01 | 3±1 | 0.2±0.1 | 0.1±0.1 | 53±11 | 40±10 | 4±3 | 0.1±0.5 | 2±2 | 0.2±0.9 | 28±29 | 72±29 | n.d. |
| 108-122 | 13 | 0.4±0.1 | 0.02±0.01 | 0.02±0.02 | 4±2 | 0.2±0.1 | 0.2±0.2 | 51±16 | 43±14 | 3±3 | 0.2±0.7 | 2±2 | 0.3±1.0 | 44±46 | 56±46 | n.d. |
| 122-200 | 18 | 0.4±0.2 | 0.02±0.01 | 0.01±0.02 | 4±1 | 0.2±0.1 | 0.1±0.2 | 52±10 | 44±9 | 2±2 | n.d. | 1±2 | 0.7±2.3 | 22±30 | 78±30 | n.d. |

**Table 2: Diapycnal fluxes and supplies (in bold) of $O_2$ and DOM: DOC, DON, dissolved organic carbon in DCCHO and DHAA and dissolved organic nitrogen in DHAA. 95% confidence intervals, calculated after Schafstall et al. (2010) for each parameter, are presented in brackets. BLM – "below the mixed layer" – a depth, defined below 10m of the water column, using a threshold criterion of 0.2°C temperature decrease.**

| | Depth | DOC | DON | DCCHO-C | DHAA-C | DHAA-N | O₂ |
|---|---|---|---|---|---|---|---|
| Flux (mmol m⁻² day⁻¹) | 10-24 | 31 (25) | -0.6 (0.4) | 6 (2) | 0.9 (0.4) | 0.3 (0.1) | 50 (14) |
| | 24-38 | 5 (17) | 8 (6) | 0.2 (0.1) | 0.07 (0.03) | 0.03 (0.01) | 32 (10) |
| | 38-52 | 0.4 (0.5) | 0.4 (0.9) | 0.12 (0.07) | 0.07 (0.03) | 0.03 (0.02) | 32 (9) |
| | 52-66 | 0.2 (0.2) | 0.5 (1.3) | 0.01 (0.01) | 0.05 (0.02) | 0.02 (0.01) | 17 (6) |
| | 66-80 | 0.6 (0.7) | 0.1 (0.3) | 0.12 (0.06) | 0.02 (0.01) | 0.7×10⁻² (0.4×10⁻²) | 8 (4) |
| | 80-94 | -0.04 (0.35) | -0.1×10⁻² (0.3×10⁻²) | 0.14 (0.07) | 0.01 (0.02) | 0.4×10⁻² (0.5×10⁻²) | 0.12 (0.07) |
| | 94-108 | -0.2 (0.2) | 0.05 (0.04) | 0.09 (0.08) | 0.6×10⁻² (0.5×10⁻²) | 0.2×10⁻² (0.1×10⁻²) | 0.016 (0.7×10⁻²) |
| | 108-122 | -0.2 (0.2) | 0.01 (0.08) | -0.01 (0.01) | 0.2×10⁻³ (0.3×10⁻²) | 0.1×10⁻³ (0.2×10⁻²) | 0.02 (0.01) |
| Supply (µmol L⁻¹ day⁻¹) | 10-38 | 1.8 (2.1) | - | 0.4 (0.2) | 0.06 (0.03) | 0.02 (0.01) | 1.2 (1.2) |
| | 24-52 | 0.3 (1.2) | 0.6 (0.4) | 0.5×10⁻³ (0.8×10⁻²) | 0.1×10⁻³ (3.2×10⁻³) | 0.2×10⁻⁴ (0.14×10⁻³) | 0.04 (0.93) |
| | 38-66 | 0.01 (0.04) | -0.01 (0.1) | 0.8×10⁻² (0.5×10⁻²) | 0.1×10⁻² (0.3×10⁻²) | 0.1×10⁻² (0.1×10⁻²) | 1.0 (0.8) |
| | 52-80 | -0.03 (0.05) | 0.03 (0.09) | -0.8×10⁻² (0.5×10⁻²) | 0.2×10⁻² (0.2×10⁻²) | 0.7×10⁻³ (0.6×10⁻³) | 0.7 (0.5) |
| | 66-94 | 0.05 (0.05) | 0.01 (0.02) | -0.1×10⁻³ (0.7×10⁻³) | 0.1×10⁻³ (0.2×10⁻³) | 0.2×10⁻³ (0.5×10⁻³) | 0.5 (0.3) |
| | 80-108 | 0.01 (0.03) | -0.3×10⁻² (0.3×10⁻²) | 0.4×10⁻³ (0.8×10⁻³) | 0.4×10⁻³ (1.4×10⁻³) | 0.1×10⁻³ (0.4×10⁻³) | 0.7×10⁻² (0.5×10⁻²) |
| | 94-122 | 0.4×10⁻³ (0.017) | 0.2×10⁻³ (0.6×10⁻³) | 0.7×10⁻³ (0.6×10⁻³) | 0.4×10⁻³ (0.3×10⁻³) | 0.1×10⁻³ (0.1×10⁻³) | -0.1×10⁻² (0.1×10⁻³) |

| | Depth (m) | DOC | DON | DCCHO-C | DHAA-C | DHAA-N | $O_2$ |
|---|---|---|---|---|---|---|---|
| **Flux (mmol $m^{-2}$ $day^{-1}$)** | BML-24 | 31 (+56/-6) | -0.6 (+0.1/-1.0) | 6 (+8/-0.06) | 0.9 (+1.3/+0.1) | 0.3 (+0.4/+0.05) | 50 (+77/+17) |
| | 24-38 | 5 (+24/-12) | 8 (+87/-2) | 0.2 (+6/-0.01) | 0.07 (+0.4/+0.03) | 0.03 (+0.15/+0.013) | 32 (+77/+11) |
| | 38-52 | 0.4 (+1.2/-0.1) | 0.4 (+8/-1) | 0.12 (+2/+0.04) | 0.07 (+0.3/+0.04) | 0.03 (+0.1/+0.01) | 32 (+72/+15) |
| | 52-66 | 0.2 (+0.6/-0.003) | 0.5 (+14/-2) | 0.01 (+16/-0.9) | 0.05 (+0.2/+0.03) | 0.02 (+0.1/+0.01) | 17 (+89/+5) |
| | 66-80 | 0.6 (+1.8/-0.03) | 0.1 (+12/-2) | 0.12 (+11/-0.5) | 0.02 (+0.5/-0.08) | $0.7 \times 10^{-2}$ (+0.2/-0.03) | 8 (+17/+1) |
| | 80-94 | -0.5 (+0.3/-0.4) | $-0.1 \times 10^{-2}$ (+0.01/-0.06) | 0.14 (+11/-0.5) | 0.01 (+0.2/-0.02) | $0.4 \times 10^{-2}$ (+0.06/-0.01) | 0.12 (+0.2/+0.03) |
| | 94-108 | -0.2 (+0.02/-0.4) | 0.05 (+11/-2) | 0.09 (+24/-1) | $0.6 \times 10^{-2}$ (+0.3/-0.05) | $0.2 \times 10^{-2}$ (+0.1/-0.02) | 0.016 (+0.04/+0.01) |
| | 108-122 | -0.2 (-0.06/-0.4) | 0.01 (+3/-0.5) | -0.01 (+0.3/-4) | $0.2 \times 10^{-3}$ (+0.01/-0.02) | $0.1 \times 10^{-3}$ (+0.01/-0.001) | 0.02 (+0.06/+0.01) |
| **Supply (µmol $kg^{-1}$ $day^{-1}$)** | BML-38 | 1.8 (+4.0/-1.0) | -0.6 (+5/-1) | 0.4 (+0.8/-0.02) | 0.06 (+0.09/+0.005) | 0.02 (+0.03/+0.002) | 1.2 (+5/-2) |
| | 24-52 | 0.3 (+1.6/-0.9) | 0.6 (+6/-0.2) | $0.5 \times 10^{-2}$ (+0.4/-0.01) | $0.1 \times 10^{-3}$ (+0.02/-0.003) | $0.2 \times 10^{-4}$ (+0.01/-0.001) | 0.04 (+4/-2) |
| | 38-66 | 0.01 (+0.07/-0.03) | -0.01 (+1/-0.2) | $0.8 \times 10^{-2}$ (+0.2/+0.002) | $0.1 \times 10^{-2}$ (+0.02/-0.001) | $0.5 \times 10^{-3}$ (+0.01/-6×10⁻⁴) | 1.0 (+7/-0.5) |
| | 52-80 | -0.03 (+0.05/-0.08) | 0.03 (+1/-0.2) | $-0.8 \times 10^{-2}$ (+1/-0.1) | $0.2 \times 10^{-2}$ (+0.04/-0.005) | $0.7 \times 10^{-3}$ (+0.01/-0.002) | 0.7 (+6/-0.3) |
| | 66-94 | 0.05 (+0.13/-0.006) | 0.01 (+0.9/-0.1) | $-0.1 \times 10^{-2}$ (+1/-0.1) | $0.6 \times 10^{-3}$ (+0.04/-0.007) | $0.2 \times 10^{-3}$ (+0.01/-0.003) | 0.5 (+1/+0.07) |
| | 80-108 | $0.8 \times 10^{-2}$ (+0.03/-0.02) | $-0.3 \times 10^{-2}$ (+0.8/-0.1) | $0.4 \times 10^{-2}$ (+2/-0.1) | $0.4 \times 10^{-3}$ (+0.02/-0.04) | $0.1 \times 10^{-3}$ (+0.007/-0.001) | $0.7 \times 10^{-2}$ (+0.01/+0.001) |
| | 94-122 | $0.4 \times 10^{-2}$ (+0.02/-0.01) | $0.2 \times 10^{-2}$ (+0.8/-0.1) | $0.7 \times 10^{-2}$ (+2/-0.3) | $0.4 \times 10^{-3}$ (+0.02/-0.004) | $0.1 \times 10^{-3}$ (+0.006/-0.001) | $-0.1 \times 10^{-2}$ (+0.003/-0.002) |

**Figure 1:** Study area and station map. CTD stations, where CTD-probe and fluorimeter measurements were accomplished are marked as black dots (a,b). PUMP-CTD stations are depicted in pink diamonds (a). CTD and PUMP-CTD stations, where DOM sampling was performed are marked as green stars (a). Microstructure measurements, combined with oxygen profiles are marked as grey circles (b). Microstructure measurements, combined with dissolved organic matter (dissolved organic carbon (DOC), dissolved hydrolysable amino acids (DHAA) and dissolved combined carbohydrates (DCCHO)) measurements marked as green pentagrams (b). Extra microstructure measurements, combined with DOC measurements marked with violet pentagrams (b). Shaded colors represent chl $a$ concentrations at upper 10 m depth (a) and oxygen concentrations at 15m depth (b). Spaces between data points were interpolated by using TriScatteredInterp function (MATLAB, MathWorks).

**Figure 2:** Mean vertical distribution of the temperature (a), salinity (b), (c) chlorophyll $a$ (chl $a$) and (d) $O_2$. $O_2$ values below 1 µmol $kg^{-1}$ are shaded in violet. The data from all transects and stations were averaged over intervals of 10 km on "Distance from the coast" axis and over 1 m on "Depth" axis. Isolines represent potential density, averaged over intervals of 10 km on "Distance from the coast" axis and over 1 m on "Depth" axis.

**Figure 3:** Dissolved organic carbon (DOC) (a), dissolved organic nitrogen (DON) (b), dissolved combined carbohydrates (DCCHO) (c) and dissolved hydrolysable amino acids (DHAA) (d) distributions over the water column. Data from all transects and stations were plotted against distance to coast (km). Space between data points was interpolated by using TriScatteredInterp function (MATLAB, MathWorks). Isolines represent potential density, averaged over intervals of 10 km on "Distance from the coast" axis and over 1 m on "Depth" axis.

**Figure 4:** Vertical distribution of $O_2$ (a), DOC (b), DON (c), DCCHO(C) (d), DHAA(C) (e), DHAA(N) (f). Black line and error bar represent mean distribution and standard deviations of the data points (grey circles), respectively. The blue and red lines and shaded areas represent the mean distributions and standard deviations of parameters onshore (<40 km) and offshore (>40 km), respectively.

**Figure 5:** The PCA analysis output: variables (on the left) and individuals scores of samples (from the right). The samples, collected above 50m depth are marked with acronym "s", the ones, below 50m depth – with acronym "d". The samples, which are used for comparison are marked with acronyms "HOT" and "BATS", and represented well oxygenated samples, collected from open Pacific and open Atlantic Oceans, respectively (Kaiser and Benner, 2009).